# Blending Supervised and Reinforcement Fine-Tuning with Prefix Sampling

## Abstract

Existing post-training techniques for large language models are broadly categorized into Supervised Fine-Tuning (SFT) and Reinforcement Fine-Tuning (RFT). Each paradigm presents a distinct trade-off: SFT excels at mimicking demonstration data but can lead to problematic generalization as a form of behavior cloning. Conversely, RFT can significantly enhance a model's performance but is prone to learn unexpected behaviors, and its performance is sensitive to the initial policy. In this paper, we propose a unified view of these methods and introduce Prefix-RFT, a hybrid approach that synergizes learning from both demonstration and exploration. Using mathematical reasoning problems as a testbed, we empirically demonstrate that Prefix-RFT is both simple and effective. Not only does it surpass the performance of standalone SFT and RFT, but it also outperforms parallel mixed-policy RFT methods. A key advantage is its seamless integration into existing open-source frameworks, requiring only minimal modifications to the standard RFT pipeline. Our analysis highlights the complementary nature of SFT and RFT, validating that Prefix-RFT effectively harmonizes these two learning paradigms. Furthermore, ablation studies confirm the method's robustness to variations in the quality and quantity of demonstration data. We hope this work offers a new perspective on LLM post-training, suggesting that a unified paradigm that judiciously integrates demonstration and exploration could be a promising direction for future research.

## 1 Introduction

LLM post-training is primarily accomplished through two distinct paradigms: supervised fine-tuning (SFT) and reinforcement fine-tuning (RFT). SFT adapts pre-trained models by continuing to train them on curated datasets of labeled examples. Its strength lies in its simplicity in training: mimic the "correct" demonstrations provided in the fine-tuning dataset. it is thus highly effective for teaching models to follow instructions (Peng et al., 2023) and perform other downstream tasks (Wei et al., 2022; Zhu et al., 2025; Huang et al., 2024). However, SFT is fundamentally a form of behavioral cloning. The approach can lead to a problematic generalization when the model adopts solution paths that may be suboptimal or distant from its distribution (Chu et al., 2025a; Chen et al., 2025a). Compounded by the exposure bias inherent in the training method, the model's robustness can be hampered, especially in complex tasks such as reasoning (Xie et al., 2024).

The emergence of reinforcement fine-tuning (RFT) has been pivotal in moving beyond the limitations of SFT and further elevating model capabilities. This next step in post-training allows a model to learn from a more dynamic and nuanced feedback signal (a.k.a. reward) than the static examples used in SFT. More recently, this paradigm has been extended to reinforcement learning from verifiable rewards (RLVR), where the reward depends on producing the correct verifiable answer (Hu et al., 2025; Xie et al., 2025). Recent large reasoning models, such as OpenAI-o1 (Jaech et al., 2024) and DeepSeek-R1 (Guo et al., 2025), demonstrated the promise of this approach. By using reinforcement learning to optimize for verifiable outcomes, these models have effectively solved problems previously considered intractable, such as competition-level math (Li et al., 2024) and coding problems (Jain et al., 2024). Despite these successes, the RFT paradigm is not uncontentious and faces its own challenges: First, the learning signal from rewards is often sparse; for complex, multi-step tasks, it is difficult to assign credit to the specific tokens that led to a successful outcome, resulting in unexpected behaviors like language mixing after training (Guo et al., 2025; Yuan et al., 2025).

Moreover, its effectiveness is highly dependent on the strength of the initial policy (Yue et al., 2025a; Zhao et al., 2025). The process arguably refines and aligns existing capabilities rather than instilling new knowledge (Liu et al., 2025d), leading some work to question whether RL can truly raise a model's intrinsic capability ceiling (Chu et al., 2025b; Liu et al., 2025a; Yue et al., 2025b; Cheng et al., 2025). The gains from RFT, while significant, may stem from perfecting what the model has already learned during pre-training and SFT (Wang et al., 2025b; Gandhi et al., 2025).

In total, SFT provides crucial dense supervision for injecting knowledge that a model cannot discover on its own. RFT, by contrast, targets actual competence but is tethered to the capabilities of the model. This establishes their core complementarity: SFT acts as the mechanism to expand the model's knowledge boundary, elevating RFT's capability ceiling, while RFT provides the goal-oriented training objective necessary to steer the model from behavioral cloning towards robust problem-solving. In practice, this synergy is already leveraged (if imperfectly), with a 'cold start' phase of SFT typically preceding RFT (Guo et al., 2025; Liu & Zhang, 2025). Yet, this is more of a heuristic than a principled strategy. The question of what defines an optimal initial policy for RFT remains open, and the interplay between the two stages and training paradigms is still to be understood. This motivates our central research question: How can we move beyond an empirical, sequential pipeline and develop a framework that formally integrates the process supervision of SFT with the goal-oriented optimization of RFT?

To bridge this gap, we first present a unified view of SFT and RFT, suggesting that they share a consistent optimization structure. We then introduce *Prefix Reinforcement Finetuning* (**Prefix-RFT**) as a hybrid post-training approach to incorporate offline demonstration datasets into the RFT training. Specifically, we sample a prefix from the demonstration and task the policy with generating its continuation. This composite sequence—an off-policy prefix followed by the on-policy continuation—is then treated as a trajectory and used alongside standard model rollouts in the RFT update step. The core intuitions behind Prefix-RFT are twofold: (1) Compared to RFT, a high-quality prefix serves as a powerful guiding mechanism for exploration. If a hybrid trajectory yields a higher reward, the corresponding prefix is naturally reinforced into the model. (2) Compared to SFT, Prefix-RFT keeps RFT's problem-solving training objective. Meanwhile, by providing only the initial part of the solution, Prefix-RFT grants the model constrained autonomy: it starts by following a promising path but still has the flexibility to discover a superior continuation, thus leveraging demonstration data for guidance without being rigidly constrained by it.

We choose math reasoning problems as the test bed for our proposed method. Despite its simplicity, our empirical results demonstrate that the Prefix-RFT outperforms naïve SFT, RFT, the two-staged SFT-then-RFT baselines, and other recent parallel works (Yan et al., 2025; Ma et al., 2025). We also validate our method across different model scales, model families, and demonstration quantities and qualities. Our further analysis reveals that Prefix-RFT enables the model to solve problems where the RFT struggles and also pushes the model to learn more from demonstration for challenging problems compared to easier ones. Taken together, our work reconsiders the view that treats SFT and RFT as two distinct and consecutive stages, suggesting that a more integrated approach to combining both learning paradigms could be a promising and valuable direction.

## 2 A UNIFIED VIEW ON SFT AND RFT TRAINING OBJECTIVES

In the mainstream LLM training pipeline, SFT and RFT are typically regarded as two distinct stages, each with its own objectives and methodologies. This section is intended to demonstrate that, despite originating from different theoretical foundations, the core dynamics of their parameter updates are inherently consistent. For simplicity, we only consider the training objective and its gradient for one data point, *i.e.*, one response in SFT or one rollout in RFT. We use $t$ to denote the token index in that data point and use the $\pi_\theta$ to represent the optimized model.

**SFT** The SFT training objective seeks to imitate high-quality expert demonstrations by minimizing the negative log-likelihood. The data point could be human expert-written demonstrations or outputs from a superior model, which we can say are sampled from an offline expert policy, $\pi_{\text{off}}$. Thus, for a model $\pi_\theta$, a prompt $x$, and a demonstration $y^* \sim \pi_{\text{off}}(\cdot|x)$, the SFT loss and its gradient are

$$L_{\text{SFT}}(\theta) = -\log \pi_\theta(y^*|x) \quad \Rightarrow \quad \nabla_\theta L_{\text{SFT}} = -\sum_t \nabla_\theta \log \pi_\theta(y_t^*|x, y_{<t}^*)$$

The gradient $\nabla_\theta L_{\text{SFT}}$ provides a low-variance signal that pushes the model $\pi_\theta$ directly towards the expert data distribution $\pi_{\text{off}}$, which is usually *not directly accessible* in practice.

**Policy Gradient** On the other hand, RFT methods use the current policy $\pi_\theta$ to generate the rollout $y \sim \pi_\theta(\cdot|x)$ and collect rewards, then use these samples to compute the policy gradient to update the policy model $\pi_\theta$ (Sutton et al., 1999). Specifically, for LLM post-training, we can treat each token generation as a separate action. Thus, the policy gradient we use to update $\pi_\theta$ is

$$\nabla_\theta L_{\text{RFT-PG}} = \sum_t \hat{A}_t \nabla_\theta \log \pi_\theta(y_t|x, y_{<t})$$

where $\hat{A}_t$ is the estimated advantage for generating the token $y_t$, and is usually the same for all tokens in the response when applying value-model free RFT algorithms like GRPO (Shao et al., 2024). The two terms essentially decides *how much* and *in which direction* to update the policy.

**PPO Training Objective** The vanilla policy gradient is strictly on-policy. To improve sample efficiency, RFT for LLMs generally employs the Proximal Policy Optimization (PPO) style objective (Schulman et al., 2017; Shao et al., 2024). The core idea is to enable multiple gradient updates with the collected samples, *i.e.*, the sample generated by the "old" policy $\pi_{\theta_{\text{old}}}(\cdot|x)$ is used to update the current policy $\pi_\theta(\cdot|x)$. Leveraging the importance sampling to correct the distribution shift and the clipping technique, the PPO training objective $L_{\text{RFT-PPO}}(\theta)$ can be expressed as:

$$L_{\text{RFT-PPO}}(\theta) = \sum_t \min\left[r_t \cdot \hat{A}_t, \text{clip}\left(r_t, 1-\epsilon, 1+\epsilon\right) \cdot \hat{A}_t\right]$$

where $r_t = \frac{\pi_\theta(y_t|x, y_{<t})}{\pi_{\theta_{\text{old}}}(y_t|x, y_{<t})}$ is the ratio between $\pi_\theta$ and $\pi_{\theta_{\text{old}}}$. Thus, its gradient is calculated as

$$\nabla_\theta L_{\text{RFT-PPO}} = \sum_t \mathbb{I}\left(\left\{\hat{A}_t > 0 \text{ and } r_t \leq 1+\epsilon\right\} \text{ or } \left\{\hat{A}_t < 0 \text{ and } r_t \geq 1-\epsilon\right\}\right) \hat{A}_t \nabla_\theta r_t(\theta)$$

$$= \sum_t \mathbb{I}_{\text{clip}}(r_t, \hat{A}_t)\hat{A}_t \nabla_\theta r_t = \sum_t \mathbb{I}_{\text{clip}}(r_t, \hat{A}_t)\hat{A}_t r_t \, \nabla_\theta \log \pi_\theta(y_t|x, y_{<t})$$

Comparing $\nabla_\theta L_{\text{SFT}}$, $\nabla_\theta L_{\text{RFT-PG}}$ and $\nabla_\theta L_{\text{RFT-PPO}}$, all methods function by applying a gradient to the log probability of a sequence, where $\nabla_\theta L_{\text{SFT}}$ updates the log probability of an expert sequence $y^*$, implicitly treating its advantage as 1. $\nabla_\theta L_{\text{RFT-PG}}$, weighted by the advantage $\hat{A}_t$, leads the model to adjust the log probability of the trial-and-error-discovered trajectories. $\nabla_\theta L_{\text{RFT-PPO}}$ takes one step further. The gradient of the log probability is multiplied by a dynamic, per-token weight $\mathbb{I}_{\text{clip}}(r_t, \hat{A}_t)\hat{A}_t r_t$ where the clipping $\mathbb{I}_{\text{clip}}(r_t, \hat{A}_t)$ penalizes large policy changes.

**Hybrid Approach** Given this inherent consistency, we propose a hybrid post-training objective for blending SFT and RFT. Consider a set of $N$ responses $\{y^{(1)}, \ldots, y^{(N)}\}$ for a prompt $x$. These responses may originate entirely from the online policy $\pi_{\theta_{\text{old}}}$, the offline expert policy $\pi_{\text{off}}$, or a composition of both. For the $i$-th response, we partition the token indices into two sets: $\mathcal{T}_{\text{exp}}^{(i)}$ for tokens generated by the model (exploration) and $\mathcal{T}_{\text{imit}}^{(i)}$ for tokens from offline demonstrations (imitation). The gradient used to optimize the policy is formulated as:

$$\nabla_\theta L_{\text{Hybrid}} = -\frac{1}{N}\sum_{i=1}^{N}\underbrace{\sum_{t \in \mathcal{T}_{\text{exp}}^{(i)}} \alpha_{i,t}\nabla_\theta \log \pi_\theta(y_t^{(i)}|x, y_{<t}^{(i)})}_{\text{learning from exploration}} + \underbrace{\sum_{t \in \mathcal{T}_{\text{imit}}^{(i)}} \beta_{i,t}\nabla_\theta \log \pi_\theta(y_t^{(i)}|x, y_{<t}^{(i)})}_{\text{learning from imitation}} \quad (1)$$

where $\alpha_{i,t}$ and $\beta_{i,t}$ represent the specific weights assigned to the $t$-th token of the $i$-th response.

# 3 PREFIX REINFORCEMENT FINE-TUNING

Our motivation stems from the complementarity between RFT and SFT learning paradigms. Specifically, the optimization goal for SFT is to push the model to fit the target distribution. When training LLMs, SFT may provide overly restricted supervision, *i.e.*, mimicking every single token. Thus, sometimes the training signal poorly aligns with the performance of downstream tasks. However, SFT still

provides reliable optimization directions and ensures the model captures accurate problem-solving patterns in the high-quality data. On the contrary, RFT uses generations from the model itself to gently carve the model's behaviors. Although promising results have been achieved, recent research raises questions about whether it could truly lift the upper bound of the base policy, indicating that the absence of explicit external guidance on the exploration may heavily restrict how far we can go with pure RFT.

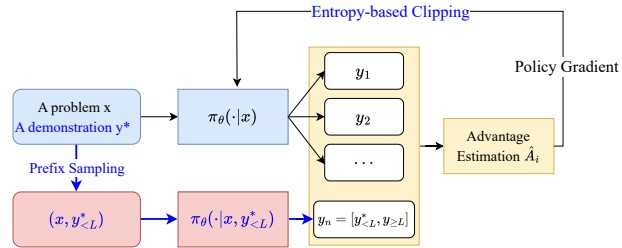

Figure 1: The Prefix-RFT does minimal modification to the existing RFT training pipeline. Given a problem and a demonstration, a prefix is sampled to guide the online continuation. The concatenated sequence $y_n$ is mixed with other online rollouts to perform RFT-style training. We also utilize an entropy-based clipping strategy to constrain the update on demonstration.

By combining them into a hybrid fine-tuning framework, we aim to benefit from SFT's stable knowledge acquisition while leveraging the exploratory power of reinforcement learning. As shown in the Fig. 1, Prefix-RFT is to *use offline demonstration prefixes $y^*_{<L}$ as guiding hints and then mix prefixes with its on-policy continuation and other on-policy rollouts to perform RFT-style training*. Given a prompt $x$ and a demonstration $y^*$, we start as in a standard RFT pipeline by generating $N-1$ online rollouts $\{y^{(1)}, \ldots, y^{(N-1)}\}$ with the current policy $\pi_{\theta_{\text{old}}}$. For the $N$-th sequence, we truncate $y^*$ to a prefix $y^*_{<L}$ and use $\pi_{\theta_{\text{old}}}$ to sample its continuation $y_{\geq L}$. We then stitch the $y^*_{<L}$ and $y_{\geq L}$ to form the hybrid trajectory $y^{(N)}$. All sequences are used to estimate advantages $\hat{A}_t$. We follow the unified view in Eq. 1 by setting the token-specific weights $\alpha_{i,t} = \beta_{i,t} = \mathbb{I}_{\text{clip}}(r_t, \hat{A}_t)\hat{A}_t r_t$, where $r_t$ is the standard PPO probability ratio. Consequently, the gradient update becomes:

$$\nabla_\theta L_{\text{PRFT}} = -\frac{1}{N} \left( \underbrace{\sum_{i=1}^{N-1} \sum_t \mathcal{W}_{i,t}^{\text{PPO}} \nabla \log \pi + \sum_{t \geq L} \mathcal{W}_{N,t}^{\text{PPO}} \nabla \log \pi}_{\text{Exploration: Standard Rollouts + Continuation}} + \underbrace{\sum_{t < L} \mathcal{W}_{N,t}^{\text{PPO}} \nabla \log \pi}_{\text{Imitation: Prefix Guidance}} \right) \quad (2)$$

where $\mathcal{W}_{i,t}^{\text{PPO}}$ denotes the clipped PPO weight $\mathbb{I}_{\text{clip}}(r_t, \hat{A}_t)\hat{A}_t r_t$. This formulation highlights that while the prefix tokens ($t < L$) originate from the offline expert, they are reinforced using the advantage estimated from the full hybrid trajectory. The intuition behind prefix sampling is to provide the model with partial guidance, allowing the policy model to explore the continuation rather than enforcing the model to mimic the entire sequence. Employing the same ratio and the clipping mechanism as in PPO, we penalize the large updates from the demonstration data. Meanwhile, the advantage assigned can indicate the value of the given prefix. Therefore, for prompts where the model does not perform well, the high-quality prefix will receive higher gradient weights, enabling the model to benefit more updates from this partial demonstration.

**Entropy-based Clipping for Constrained Update on Demonstrations** In practice, $\pi_{\text{off}}$ may be far from the current policy. The probability $\pi_\theta$ of offline tokens is therefore generally low. In this case, the gradient of demonstrations could be significantly larger than the RFT gradients (as illustrated in Table 6), potentially dominating the optimization process and preventing the model from learning via RFT. To this end, we propose an entropy-based clipping approach, *i.e.*, only involving the top-k% high-entropy demonstration tokens in the gradient calculation. Regarding implementation, we directly set the corresponding advantages of all other tokens to zero, thereby removing their contribution to the gradient. Our intuition behind the proposed strategy is twofold: First, a high entropy suggests that the output logits are relatively flat. Thus, only updating with these high-entropy tokens helps avoid too sharp distribution shifts and very large gradients. Second, a high entropy also indicates that the current policy $\pi_\theta$ is uncertain about the next generated token; these tokens could serve as critical junctures where the model is likely to deviate from the expected behavior and where reinforcement is needed (Wang et al., 2025a).

**Controlling Prefix Length with Cosine Decay Scheduler** In practice, we use a variable $l \in [0, 1]$ to determine the prefix length as $L = \lfloor l \cdot |y^*| \rfloor$. If $l \sim U(0, 1)$, the model naturally has a higher

Table 1: Main experiment results on math and general reasoning benchmarks based on **Qwen2.5-Math-7B**. **Bold** and underline indicate the best and second-best results, respectively.

| Model | Math Reasoning Performance | | | | | | General Domain Reasoning Performance | | | |
|---|---|---|---|---|---|---|---|---|---|---|
| | AIME 24/25 | AMC | MATH-500 | Minerva | Olympiad | Avg. | ARC-c | GPQA* | MMLU-Pro | Avg. |
| Qwen2.5-Math-7B | 11.5/4.9 | 31.3 | 43.6 | 7.4 | 15.6 | 19.0 | 18.2 | 11.1 | 16.9 | 15.4 |
| *Previous RFT Results* | | | | | | | | | | |
| SimpleRL-Zero | 27.0/6.8 | 54.9 | 76.0 | 25.0 | 34.7 | 37.4 | 30.2 | 23.2 | 34.5 | 29.3 |
| Oat-Zero | **33.4**/11.9 | 61.2 | 78.0 | 34.6 | 43.4 | 43.7 | 70.1 | 23.7 | 41.7 | 45.2 |
| *Baselines Using the Same Dataset and Base Model* | | | | | | | | | | |
| RFT | 25.1/15.3 | 62.0 | 84.4 | 39.3 | 46.8 | 45.5 | 82.3 | **40.4** | 49.3 | 57.3 |
| SFT | 22.2/22.3 | 52.8 | 82.6 | **40.8** | 43.7 | 44.1 | 75.2 | 24.7 | 42.7 | 47.5 |
| RL w/ SFT Loss | 19.5/16.4 | 49.7 | 80.4 | 34.9 | 39.4 | 40.1 | 71.2 | 23.7 | 43.2 | 46.0 |
| SFT+RFT | 25.8/23.1 | 62.7 | 87.2 | 39.7 | 50.4 | 48.2 | 72.4 | 24.2 | 37.7 | 44.8 |
| UFT | 20.8/16.5 | 58.8 | 83.8 | 33.8 | 51.6 | 44.2 | 83.4 | 34.5 | 49.4 | 55.8 |
| ReLIFT | 28.2/20.1 | 64.9 | 87.4 | 33.8 | 52.5 | 47.8 | 76.2 | 37.9 | 52.5 | 55.5 |
| LUFFY | 29.4/23.1 | 65.6 | 87.6 | 37.5 | **57.2** | 50.1 | 80.5 | 39.9 | **53.0** | 57.8 |
| **Our Method** | | | | | | | | | | |
| Prefix-RFT | 31.8/**26.4** | **68.2** | **88.4** | 40.3 | 55.7 | **51.8** | **84.0** | 39.1 | 52.1 | **58.4** |

chance of accessing early tokens in the demonstration. However, specific skills, such as drawing conclusions or summarizing, are usually located at the end of the sequence. To alleviate this position bias, we propose using a cosine decay scheduler to control the prefix length. Specifically, the length variable $l$ is randomly sampled from $U(low, high)$, where $high$ is a constant and $low$ decreases from $high$ to near zero throughout the entire training. The design not only mitigates the position bias issue but also aligns with the existing standard SFT-and-then-RFT recipe. It also naturally introduces the curriculum learning schedule into the training.

**Parallel Works**   Note that several parallel works share a similar motivation to incorporate the offline dataset into RFT training. LUFFY (Yan et al., 2025) mixes the entire offline data with other on-policy rollouts to perform RFT-style training. More similar to us, UFT (Liu et al., 2025b) first samples a prefix from the demonstration, then uses SFT loss on the prefix with a static small weight and RFT loss on the on-policy continuations. ReLIFT (Ma et al., 2025) incorporates a staged method to interleave SFT and RFT, with the SFT focusing on challenging problems that RFT cannot solve. Compared with these methods, our approach is distinguished by its practicability, simplicity, and ease of integration into existing RFT pipelines. More discussions of related works and a detailed comparison with these parallel works can be found in Appendix A.1.

## 4 MAIN EXPERIMENTS

**Experiment Settings and Baselines**   We employ math reasoning as the test playground due to access to reliable and inexpensive verifiers. Our training data is a length-filtered subset Yan et al. (2025) of the OpenR1-Math-220K dataset (Face, 2025), comprising approximately 46k problems, each problem equipped with a demonstration generated by DeepSeek-R1. We use Qwen2.5-Math-7B (Yang et al., 2024) as our base model. The evaluation and other training details are included in the Appendix A.2. We compare with the following baselines: (1) Previous RFT recipes, including *Simple-RL* (Zeng et al., 2025) and *Oat-Zero* (Liu et al., 2025d). (2) For a fair comparison, all the following baselines use the same base model and the same dataset. The difference solely lies in how to incorporate offline data points. These baselines include *RFT*, *SFT*, *RFT w/ SFT Loss* that directly employs SFT loss on the off-policy data during RFT training, *SFT + RL* that continues RFT training with SFTed model. Concurrent work *ReLIFT, UFT,* and *LUFFY* are also included as our baselines.

**Main Results**   The results are shown in the Tab. 1. Our observations and conclusions are as follows: (1) The pure *RFT* baseline in our setting already achieves strong performance compared to established Zero-RL results, validating that the setting and comparison of our experiments are solid. (2) In our setting, the *RFT* and *SFT* baselines achieve similar performance, while *SFT+RFT* is significantly better. In particular, *SFT* contributes more on most challenging benchmarks (*i.e.* AIME25), *RFT*

benefits more on datasets where the base model already has moderate performance (*i.e.* AMC and MATH500), and *SFT+RFT* achieves a better balance, highlighting our motivation that those two learning paradigms could complement each other and could be better blended. (3) The joint training method *RL /w SFT Loss* is counterproductive. (4) Though utilizing less demonstration data, the multi-staged method like *ReLIFT* does not necessarily achieve better performance than the simple two-staged *SFT+RFT* baseline. This may be because the interleaved method requires more careful hyperparameter tuning to ensure the performance. This also highlights the need for a hybrid approach to stably exploit the offline dataset during RFT. (5) Despite its simplicity, Prefix-RFT performs well on six math reasoning benchmarks and three general domain reasoning tasks, significantly outperforming all baselines and achieving comparable performance with concurrent work *LUFFY*. We also experiment with a smaller model, Qwen2.5-Math-1.5B, and a base model from a different family, LLaMA-3.1-8B. The results clearly indicate the superior performance of our method. To highlight, our method achieves 41.0 with the Qwen 1.5B model (14.5 with Llama 8B) compared with 38.0 (13.2) from the LUFFY method. Dailed results can be found in Appendix A.3.

## 5 ANALYSIS

To better validate the underlying mechanisms of our proposed method, we investigate the following two questions: (1) *Does Prefix-RFT effectively synthesize the distinct training paradigms of SFT and RFT?* (2) *Does Prefix-RFT dynamically adjust its learning strategy during different training stages and when faced with problems of varying difficulty?* The detailed settings for our analysis experiments are as follows. Most analysis experiments are based on Qwen2.5-Math-1.5B. We employ Deepseek-Distill-r1-1.5B as $\pi_{\text{off}}$ to generate eight responses per problem. To perform multi-epoch RFT training with the limited computation budget, we sample 16k problems from the original dataset as the new training set $\text{Train}_{16k}$. We ensure that every prompt has at least one correct demonstration. We use a batch size of 128 to perform 640 training steps, resulting in 5 epochs for each method. We use the learning rate of 5e-5 for SFT and 1e-6 for RFT and Prefix-RFT . All different analysis metrics are calculated with a 2k subset from $\text{Train}_{16k}$, noted as $\text{Train}_{2k}$. To study the model's behavior on problems that the pure RFT struggles to solve, we use checkpoints from another RFT run to identify 256 problems and note this subset as $\text{Train}_{\text{hard}}$.

### 5.1 DOES PREFIX-RFT BRIDGE SFT AND RFT?

To answer this question, we calculate three metrics to measure the model's state during the training with $\text{Train}_{2k}$: (1) the Avg@16 score, which estimates the model's task performance; (2) the Best@16 score, which estimates the model's problem-solving potential after training; and (3) the SFT loss on the provided demonstrations that estimate the distribution gaps between the model and $\pi_{\text{off}}$. These results are summarized in the Fig. 2. And our key observations are as follows.

**Comparing SFT and RFT** As mentioned above, SFT and RFT present distinct training paradigms, with the former focusing on minimizing the likelihood of the given demonstration and the latter directly optimizing for task performance. Our results suggest that (1) The SFT could initially damage the model's performance and then rebuild it (the performance of the second checkpoint is lower than that of the initial one). Mimicking external demonstrations, the SFT-ed model has the potential to find the solution path for nearly all training problems (best@16 of 0.96), but cannot robustly solve them (avg@16 of 0.52). (2) On the contrary, our results indicate that RFT model achieves better overall performance (0.61 for RFT v.s. 0.52 for SFT regarding avg@16) but can be limited by the ability of the initial policy (0.85 for SFT v.s. 0.14 for RFT regarding best@16 on $\text{Train}_{\text{hard}}$). Furthermore, the best@16 converges much faster than avg@16, implying RFT admits a discover-and-gradually-reinforce learning strategy. (3) During the RFT training, the model keeps deviating from the demonstration distribution (loss on demonstration increased from 0.67 to 0.81), proving that SFT's training objective—loss on demonstration—can sometimes be a poor predictor of the exact task performance. In total, all these observations reflect the pros and cons of the two method and their complementarity, justifying our motivation for blending both learning paradigms.

**Prefix-RFT makes the best of both worlds (to some extent)** Our results demonstrate the superiority of Prefix-RFT . Compared to RFT, Prefix-RFT not only achieves higher avg@16, best@16, and a lower loss on demonstration, but also shows notable performance gains on problems previously intractable for RFT. Notably, our approach effectively aligns with the offline expert distribution,

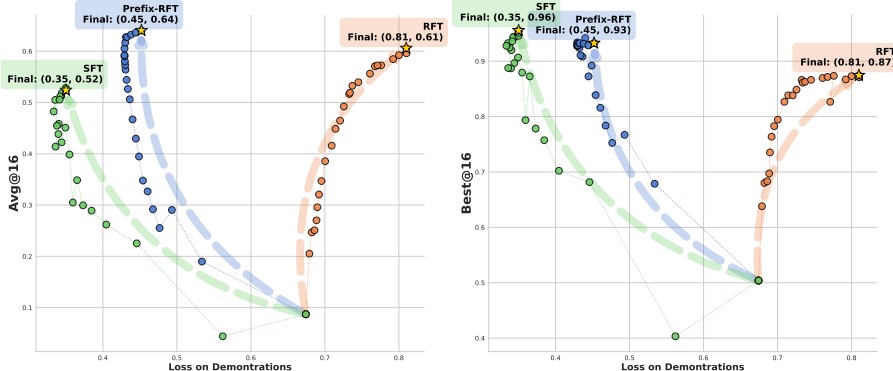

(a) Training trajectories on the Train$_{2k}$. The figure highlights the distinct learning objectives and paradigms of SFT and RFT, and indicates that Prefix-RFT effectively blends both methods regarding training objectives.

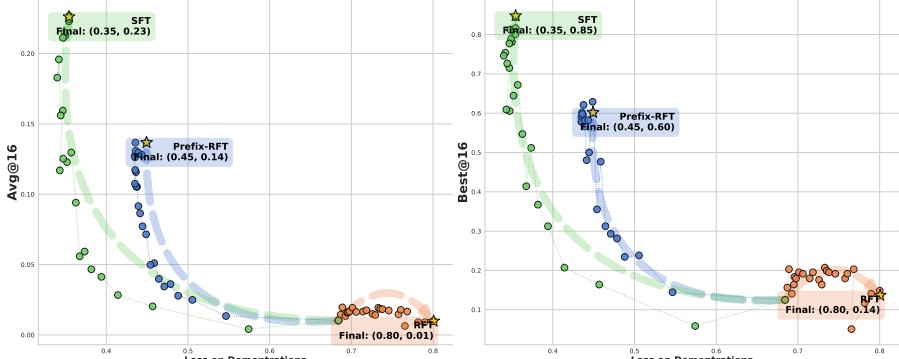

(b) Training trajectories on the Train$_{hard}$. It shows that the RFT method keeps struggling with these unsolvable problems during the training, and Prefix-RFT can achieve higher scores, effectively elevating the upper bound of the RFT tuning.

Figure 2: Training trajectories of SFT, RFT, and Prefix-RFT. The x-axis denotes the SFT loss on the demonstrations. The y-axis represents the Avg@16 and Best@16 scores. The final step is marked with the yellow star, annotated using the final SFT loss and the final score.

despite only fine-tuning on the top 20% of high-entropy tokens. We posit that the observed gap in final loss between Prefix-RFT (0.45) and SFT (0.35) is likely attributable to variances in hyperparameter settings rather than fundamental methodological differences. To investigate this, we conducted preliminary tests on hyperparameter sensitivity. For instance, training SFT with a lower learning rate of 1e-6 results in a final loss of approximately 0.42. Conversely, applying a higher learning rate of 5e-5 to RFT leads to training instability and eventual model collapse. These findings underscore that identifying optimal hyperparameters presents a significant challenge for unified training frameworks, which we designate as an avenue for future work. Furthermore, a performance gap persists between Prefix-RFT and SFT on the Train$_{hard}$ dataset, indicating that there is still space for improvement.

## 5.2 ADVANTAGE-DRIVEN UPDATES INDUCE DYNAMIC TRANSITION BETWEEN SFT AND RFT

As discussed in Sec. 3, the advantage assigned to a hybrid sequence serves as a proxy for its prefix's utility. This advantage dynamically adjusts the prefix's influence on the training update, thereby inducing a transition between SFT and RFT. We find that this transition manifests at both the level of overall training dynamics and the level of individual examples. In this subsection, we first illustrate the evolution of prefix advantages throughout the training process and then examine the relationship between the loss on demonstrations and problem difficulty.

**Overall training level** Fig. 3a shows the training dynamics of the Qwen2.5-Math-1.5B and Qwen2.5-Math-7B. The graphs plot the average reward of rollouts initiated with a prefix and the overall training reward. According to the definition of advantage calculation in GRPO algorithms, the shaded area between these two curves roughly represents the accumulated advantage assigned to the prefix. Here are our observations: (1) Both models demonstrate a remarkable ability to quickly leverage the provided prefix. The "Reward with Prefix" score surges in the initial phase, with the 7B

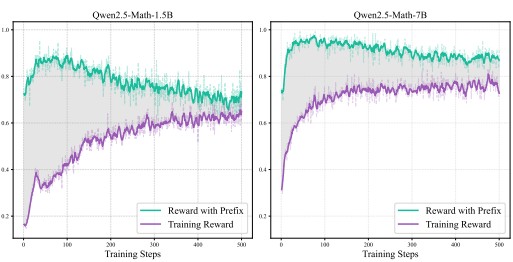 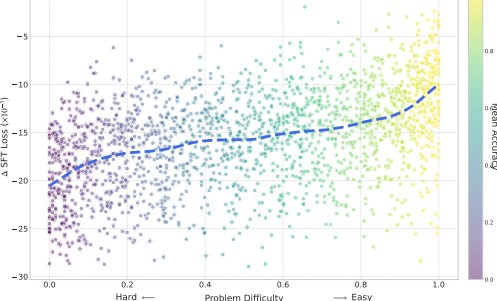

(a) Average reward of rollouts with a prefix and the overall reward. The shaded area represents the advantage assigned to the prefix. As the advantage diminishes, the training gradually transitions from SFT to RFT.

(b) The change in SFT loss on problems of varying difficulties, suggesting Prefix-RFT provides more supervision for more challenging problems.

Figure 3: Our analysis reveals that the advantage-driven update effectively induces desirable transitions between SFT and RFT, at both the level of overall training dynamics and individual examples.

model approaching a near-perfect score of 1.0 within the first 100 training steps. (2) Owing to the cosine decay scheduler, the average reward with the prefix slightly decreases. The behavior is clearer for the 1.5B model, suggesting that the smaller model is more sensitive to changes in prefix length. (3) The gap between the two average rewards stays positive and gradually narrows down through the training. This diminishing advantage signifies that the model's reliance on the prefix decreases as its own generative reasoning capabilities improve. *From the perspective of the gradient, this trend reflects a smooth and desirable transition from SFT to RFT during the training process.*

**Individual example level**   We then investigate whether such a transition also exists at the example level. We analyze the change in SFT loss on demonstration for problems of varying difficulty. This analysis focuses on the training interval from the first epoch to the third epoch, a phase chosen to bypass the initial rapid convergence and the subsequent performance saturation in later stages. Figure 3b presents the primary results of this analysis. Each point in the scatter plot corresponds to a unique problem instance, plotting its difficulty against the observed SFT loss change. Problem difficulty (x-axis) is quantified as the model's mean solution accuracy, evaluated with the multiple saved checkpoints from the 2nd epoch to the 3rd epoch. A lower accuracy thus indicates a more challenging problem. The y-axis represents the change in SFT loss on the provided demonstrations, calculated as $\Delta_{\text{loss}} = \text{Loss@384} - \text{Loss@128}$. A more negative value signifies more learning pressure from demonstration loss. The LOWESS-fitted trend line reveals a clear positive correlation: as mean accuracy increases, the change in SFT loss becomes less negative. This observation indicates that the model achieves a substantially larger loss reduction—and thus learns more intensively from the demonstrations—for problems it finds more challenging (i.e., those with lower accuracy). Conversely, for easier problems where the model already achieves high accuracy, the SFT loss reduction is marginal. This suggests that the model relies less on the demonstrations and more on its own problem-solving abilities. *This finding elucidates a key mechanism of our approach: it facilitates a dynamic, example-level transition between reliance on demonstrations and self-exploration.*

## 6 ABLATION STUDIES

**Entropy-based Clipping**   To validate the effect of entropy-based clipping, we compare five different approaches. Our primary method, labeled *top 20%*, updates the model using only the 20% of prefix tokens with the highest entropy. We compare this against four variants. Two of these, *top 50%* and *top 80%*, maintain the high-entropy selection strategy but relax the clipping ratio to 50% and 80%, respectively. The other two variants serve as controls: *random 20%* selects tokens randomly, while *bottom 20%* selects the 20% of tokens with the lowest entropy. All variants were trained for 300 steps, during which we monitored training response length, training reward, and benchmark performance.

The results clearly demonstrate the superiority of the top 20% strategy. Regarding benchmark performance, the top 20% variant exhibits a stable upward trend, achieving the highest score of approximately 50% after 300 steps and the highest training reward. Its superior performance was achieved while generating the shortest training response length (2k–2.5k tokens). In contrast, relaxing the clipping ratio ( top 50% or 80%) or altering the selection method (random or bottom) leads to diminished performance and greater instability. These results confirm our core intuition about the clipping strategy: (1) *The update on demonstrations should be constrained*: When an offline dataset

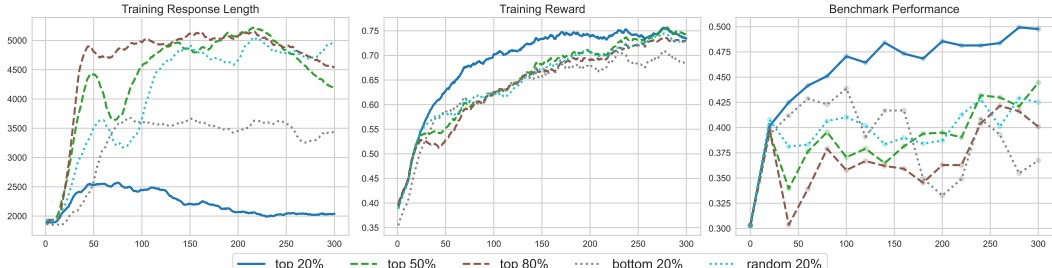

Figure 4: Ablation study results on the entropy-based clipping strategy. From left to right: training response length, training reward, and benchmark performance. Our proposed strategy, top 20%, consistently outperforms other variants.

is significantly off-policy, the gradients from the prefix tokens can overwhelm those from on-policy tokens. This can cause the training to degenerate into simple SFT on the prefix data. This phenomenon is evident in the top 80% variant, which quickly overfits to the superficial feature of response length from the demonstrations rather than optimizing for task performance. This finding aligns with our preliminary experiments on using multiple prefixes (sampled from the same demonstration) for a single problem (similar to UFT). The hybrid approach becomes counterproductive in this case, as the model struggles to balance both learning signals. (2) *High-entropy tokens provide richer learning signal*: Merely constraining the update ratio is insufficient; the strategy for selecting tokens is crucial. As shown in the figure, the *random 20%* is only a delaying tactic and is ineffective at preventing the policy from overfitting to the demonstrations. The *bottom 20%* strategy was even worse, proving detrimental to both training reward and benchmark performance. This confirms that focusing on high-entropy tokens is essential for effective learning.

**Decay Scheduler**    To investigate the effect of the proposed cosine decay scheduler, we conduct an ablation study comparing it against a *Uniform Scheduler* baseline: sampling the variable $l$ uniformly from the $[0.05, 0.95]$ throughout the entire training. The experiments are performed on both the Qwen2.5-Math-7B and 1.5B models. As illustrated in Fig. 5 (left), using the *Uniform Scheduler* results in performance degradation across all benchmarks. Beyond mitigating potential positional bias, our proposed cosine decay scheduler also more effectively modulates the training dynamics. As

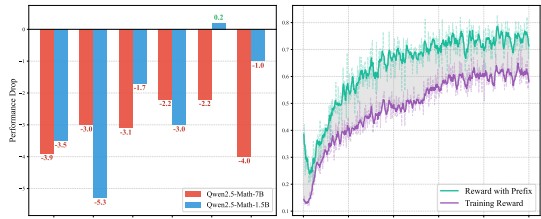

Figure 5: Effect of the scheduling strategy on benchmark performance and training dynamics. Left: Performance comparison between the proposed cosine decay scheduler and the baseline Uniform Scheduler. Right: Training reward dynamics when employing the Uniform Scheduler.

shown in Fig. 5 (right), under the uniform schedule, the gap between the prefix-initiated and the overall reward is initially small and gradually widens, which is in contrast to the pattern observed in Fig. 3a, suggesting that our cosine decay scheduler better incentivizes the model to learn from the demonstration, particularly during the initial training phase.

**Results on more models**    In addition to Qwen2.5-Math-7B, we also test our method on Qwen2.5-Math-1.5B, LLaMA-3.1-8B, and Qwen3-1.7B-Base. The training settings for LLaMA models follow exactly (Yan et al., 2025). Our baselines include SFT, RFT, LUFFY, and ReLIFT. The performance on the six math reasoning problems is summarized in Tab. 2. The results clearly indicate that the Prefix-RFT achieves superior performance on both models, regardless of their distinct architectures, scales, and initial capabilities. On the Qwen2.5-Math-1.5B model, Prefix-RFT achieved an average score of 41.1, significantly outperforming the next-best method, LUFFY, as well as the conventional SFT and RFT methods. A similar trend was observed on the LLaMA-3.1-8B and Qwen3-1.7B-base models, validating its effectiveness and robustness.

**Data-limited Scenarios**    Compared with RFT, Prefix-RFT requires extra demonstrations. Since acquiring such data can be prohibitively expensive, either from human experts or a superior model, we investigate the method's performance under two data-constrained scenarios to assess its real-world viability: (1) limited demonstration quantity, using 10% (4.5k) and 1% (0.45k) of the training data; and (2) suboptimal demonstration quality, with demonstrations generated by DeepseekR1 distillation series models of varying sizes (1.5B to 32B). All experiments use the Qwen2.5-Math-1.5B model, with results detailed in Table 3. The analysis shows that even under these strict constraints, all variants of Prefix-RFT still significantly outperform the SFT and RFT baselines. Regarding data quantity,

Table 2: Performance on Qwen2.5-Math-1.5B and LLaMa-3.1-8B as base model

| Model | AIME 24/25 | AMC | MATH-500 | Minerva | Olympiad | Avg. |
|---|---|---|---|---|---|---|
| **Qwen2.5-Math-1.5B-Base** | | | | | | |
| SFT | 11.7/13.2 | 37.8 | 70.6 | 26.8 | 31.3 | 31.9 |
| RFT | 11.8/7.7 | 40.2 | 61.8 | 26.8 | 32.0 | 30.0 |
| ReLIFT | 14.3/10.0 | 40.9 | 76.4 | 25.2 | 39.6 | 34.4 |
| LUFFY | 16.0/13.1 | 47.1 | 80.2 | 30.5 | 41.0 | 38.0 |
| Prefix-RFT | **17.7/17.7** | **50.5** | **81.4** | **32.7** | **46.5** | **41.1** |
| **LLaMA-3.1-8B-Base** | | | | | | |
| SFT | 0.5/0.1 | 5.4 | 20.2 | 4.0 | 5.3 | 5.9 |
| LUFFY | **1.9**/0.1 | **13.5** | 39.0 | 15.1 | 9.6 | 13.2 |
| ReLIFT | 1.3/0.2 | 11.9 | 35.2 | - | 11.0 | - |
| Prefix-RFT | 1.3/**1.5** | 13.3 | **40.6** | **18.1** | **11.9** | **14.5** |
| **Qwen3-Math-1.7B-Base** | | | | | | |
| SFT | 10.0/**10.4** | 34.5 | 66.6 | 24.6 | 27.2 | 28.9 |
| RFT | **11.6**/4.5 | 35.4 | 68.4 | 30.1 | 31.9 | 30.3 |
| Prefix-RFT | 10.0/8.1 | **36.3** | **74.2** | **32.7** | **35.3** | **32.8** |

Table 3: Ablation study for Prefix-RFT on the Qwen2.5-Math-1.5B model. Results show that Prefix-RFT substantially outperforms SFT and RFT baselines while demonstrating strong data efficiency and remarkable robustness to the quality of demonstration generators.

| | AIME 24/25 | AMC | MATH-500 | Minerva | Olympiad | Avg. |
|---|---|---|---|---|---|---|
| SFT | 11.7/13.2 | 37.8 | 70.6 | 26.8 | 31.3 | 31.9 |
| RFT | 11.8/7.7 | 40.2 | 61.8 | 26.8 | 32.0 | 30.0 |
| Prefix-RFT | 17.7/17.1 | 50.5 | 81.4 | 32.7 | 46.5 | 41.1 |
| *Ablations on different demonstration sizes* | | | | | | |
| Data size 4.5k | 17.8/15.9 | 49.7 | 79.0 | 35.3 | 46.8 | 40.8 |
| Data size 0.45k | 15.2/11.8 | 46.3 | 76.0 | 33.5 | 42.8 | 37.6 |
| *Ablations on different demonstration generators* | | | | | | |
| Deepseek-R1-distill-32B | 18.1/15.3 | 50.9 | 81.2 | 34.2 | 43.7 | 40.6 |
| Deepseek-R1-distill-7B | 18.1/15.9 | 49 | 79.8 | 36.4 | 44.9 | 40.7 |
| Deepseek-R1-distill-1.5B | 15.9/12.6 | 47.7 | 79 | 37.1 | 46.2 | 39.8 |

reducing the training set by 99% (from 45k to 0.45k samples) results in only a moderate performance drop (40.8 to 37.6), highlighting its data efficiency. The method also shows remarkable robustness to demonstration quality, as performance is nearly identical when using a 1.5B generator versus a 32B one. We note, however, that the most challenging benchmarks, such as AIME, are the most affected by these limitations, indicating that high-quality, large-scale demonstration data remains beneficial for tackling top-difficulty problems.

# 7 CONCLUSION

Motivated by the complementarity of SFT and RFT learning paradigms, this work presents Prefix-RFT to blend them via sampling prefixes from offline demonstrations as hints. Prefix-RFT is simple yet effective, outperforming the naive SFT, RFT, the two-staged SFT-then-RFT, and other parallel works. The method is further validated across different model scales, families, and varying demonstration quantities and qualities to indicate its robustness. Further analysis highlights that Prefix-RFT effectively guides the model to solve problems that are unlearnable to pure RFT, striking a sweet spot between RFT (by providing supervision where it's most needed) and SFT (by incorporating a goal-oriented training objective). We argue that a hybrid post-training approach is crucial for training more powerful, accessible, and agentic models, as it enables learning from a broader range of data sources. This work represents an initial endeavor, proposing a foundational framework that we hope will be elaborated upon in future research.

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

# A  APPENDIX

## A.1  RELATED WORKS

**Incorporating offline dataset for online RL**  Offline RL aims to learning a reward maximizing policy from a fixed, static dataset, collected by some existing policy (Levine et al., 2020; Lange et al., 2012). Due to the dataset limitations, offline RL often results in a suboptimal policy, motivating recent work to combine offline and online RL (Luo et al., 2023; Ball et al., 2023; Song et al., 2023; Liu et al., 2025c). In the domain of Large Language Model, the most common approach is to employ the two-stage SFT-and-then-RFT method, where SFT instills desirable patterns or skills into the model and RFT amplifies them (Liu et al., 2025a). Built upon this two-staged strategy, recent works like DR-PO (Chang et al., 2024) also investigates starting online RL from a sampled offline state (a truncated sentence for LLM) for RLHF. However, the interplay between SFT and RFT remains to be understood and may be specific to each use case (Cai et al., 2025; Chen et al., 2025a), and determining the optimal strategy to stitch the two methods remains an open question (Chen et al., 2025b). Therefore, there is an emerging body of work focusing on how to better integrate these two learning paradigms and how to incorporate the offline dataset to improve the LLM post-training (Yan et al., 2025; Liu et al., 2025b; Ma et al., 2025).

**Detailed comparison with parallel works**  Here we discuss the differences between our method and the contemporaneous works UFT, LUFFY, and ReLIFT. (1) **UFT** (Liu et al., 2025b) shares the strategy of sampling prefixes, it applies a static, pre-assigned scalar weight (0.001) to the prefix tokens. Our method, conversely, employs a dynamic weight determined by the estimated advantage of the full hybrid sequence. This allows the model to learn more intensively from prefixes that actually lead to high-value outcomes. Though theoretically justifed, UFT mainly experiment with more simpler setting, *i.e.*, smaller models and simpler offline demotnrations. Our empirical results demonstrate that our dynamic approach significantly outperforms the static weighting scheme in complex reasoning tasks. (2) **LUFFY** (Yan et al., 2025) mixes entire offline traces with on-policy data and utilizes a policy reshaping function $f(\pi_\theta) = \frac{\pi_\theta}{\pi_\theta + \lambda}$ to handle off-policy distribution shift. This method introduces a hyperparameter $\lambda$ that can be not straightforward and sensitive to tune. Our approach addresses the off-policy nature of demonstrations through a simpler entropy-based clipping mechanism, which we show to be robust and effective across different models and scales. (3) **ReLIFT** (Ma et al., 2025) adopts a staged method to interleave SFT and RFT, where SFT is selectively applied to problems that RFT fails to solve. While effective, this requires managing a multi-stage pipeline and identifying "hard" problems iteratively. In contrast, our method seamlessly blends SFT and RFT in a single training stage by using prefixes to guide exploration on difficult problems dynamically, offering a simpler and more integrated pipeline.

## A.2 EVALUATION AND EXPERIMENT HYPERPARAMETERS

Regarding **evaluation**, we follow Yan et al. (2025) and evaluate our approach with six math reasoning tasks, *i.e.*, AIME 2024, AIME 2025 (Li et al., 2024), AMC (He et al., 2024), Minerva Lewkowycz et al. (2022), OlympiadBench He et al. (2024), and MATH-500 Hendrycks et al. (2021). Because AIME 2024, AIME 2025, and AMC have fewer data points, we report avg@32, and for the other three benchmarks, we report pass@1. To see whether the reasoning learned can be generalized to other general reasoning problems, we test the model with ARC-c (Clark et al., 2018), GPQA-diamond (Rein et al., 2023), and MMLU-Pro (Wang et al., 2024).

**Hyperparameters**  Most of our training hyperparameters are set as Yan et al. (2025) to ensure fair comparison. Regarding our method-specific hyperparameters, unless specified otherwise, we sample 8 rollouts per prompt, and one of them starts with the sampled prefix. And for each mini-batch, we only update the top 20% prefix tokens that are high-entropy. The model is trained for 500 steps. And each time step $t$, we sample $l$ uniformly from $[\text{low}_t, 0.95]$ to decide the prefix length as $l$ times the total demonstration length. And $\text{low}_t$ follows a cosine decay scheduler, starting from 0.95 and decaying to 0.05 at the 500th step. We use Dr.GRPO (Liu et al., 2025d) as our RFT algorithm.

## A.3 MORE EXPERIMENT RESULTS

**Performance with Large Sampling Budgets**  To investigate whether Prefix-RFT expands the model's reasoning boundaries beyond simply improving the likelihood of known solutions, we evaluate the model's performance using a large sampling budget ($n = 2048$). We compare the Base model (Qwen2.5-Math-7B-Base), SFT (checkpoint from LUFFY paper), RFT, and Prefix-RFT on the challenging AIME 2024 and AIME 2025 benchmarks. As shown in Table 4, Prefix-RFT consistently outperforms both SFT and RFT across all sampling budgets ($k = 1$ to $k = 2048$). Notably, on AIME 2025, Prefix-RFT achieves a Pass@2048 of 76.7%, significantly higher than the Base model (70.0%) and RFT (70.0%), demonstrating that our method effectively elevates the upper bound of the model's reasoning capabilities.

Table 4: Pass@$k$ performance on AIME 2024 and AIME 2025 with a sampling budget of $n = 2048$.

| Method | p@1 | p@2 | p@4 | p@8 | p@16 | p@32 | p@64 | p@128 | p@256 | p@512 | p@1024 | p@2048 |
|---|---|---|---|---|---|---|---|---|---|---|---|---|
| **AIME 2024** | | | | | | | | | | | | |
| base model | 15.22 | 23.62 | 32.49 | 40.59 | 48.00 | 55.03 | 61.79 | 68.11 | 73.88 | 79.21 | 83.59 | 86.67 |
| SFT | 25.05 | 33.78 | 43.60 | 53.18 | 60.95 | 66.45 | 69.67 | 71.43 | 72.93 | 74.67 | 76.20 | 76.67 |
| RFT | 27.29 | 34.42 | 41.83 | 48.91 | 55.11 | 60.23 | 64.59 | 68.98 | 73.90 | 79.48 | 85.42 | **90.00** |
| **Prefix-RFT** | **31.24** | **40.08** | **48.93** | **56.90** | **63.24** | **67.94** | **71.64** | **75.47** | **79.92** | **84.05** | **87.28** | **90.00** |
| **AIME 2025** | | | | | | | | | | | | |
| base model | 7.13 | 11.42 | 16.68 | 22.56 | 28.89 | 35.31 | 41.64 | 48.32 | 55.61 | 62.36 | 67.27 | 70.00 |
| SFT | 22.66 | 29.02 | 34.60 | 39.70 | 44.74 | 49.52 | 53.71 | 57.32 | 60.47 | 63.10 | 64.89 | 66.67 |
| RFT | 12.15 | 14.97 | 18.43 | 23.31 | 30.36 | 39.29 | 48.31 | 56.01 | 61.63 | 64.94 | 67.45 | 70.00 |
| **Prefix-RFT** | **25.98** | **29.77** | **34.68** | **40.07** | **45.37** | **50.78** | **56.75** | **62.74** | **67.80** | **71.95** | **75.39** | **76.67** |

**More Ablation Studies**  We further investigate the specific contributions of our design choices by comparing Prefix-RFT with several variants. We categorize these ablations into two groups: (1) *Prefix Gradient Strategies* and (2) *Methodological Variants*. All experiments are conducted on Qwen2.5-Math-7B. We aim to answer the following questions?

**Do we need to update the prefix?**  We compare *Freeze Prefix* (gradients from prefix tokensa are all clipped) and *Update All* (no clipping on prefix). As shown in Table 5, freezing the prefix results in performance similar to standard RFT (45.4 vs 45.5), indicating that explicit learning from the demonstration is crucial. Conversely, updating all prefix tokens without clipping (*Update All*) leads to only marginal gains (45.7) and undesirable training dynamics (*i.e.*, rollout length explosion), confirming the necessity of our entropy-based constraint.

**Is dynamic weighting necessary?**  We replace our entropy clipping with a *Static Weight* strategy (applying a constant 0.001 weight to prefix tokens, similar to UFT). This variant yields an average score of 43.8, significantly underperforming our dynamic approach (51.8). This supports our hypoth-

esis that weighting updates by the hybrid trajectory's advantage allows the model to selectively learn from high-value prefixes.

**Is the gain just from clipping?** To prove that our gains are not solely due to the entropy clipping technique itself, we apply our "top-20% clipping" strategy to the on-policy rollouts of a standard RFT run (*RFT + On-policy Clip*). This achieves 43.8, performs similarly to naive RFT baseline. This confirms that the performance leap is driven by the *learning from prefix*, not the clipping trick itself.

**Comparison with DR-PO Variant:** Finally, we evaluate a variant inspired by DR-PO, where we initialize rollouts from prefixes but do not include them in the loss calculation. The difference between DR-PO variant and prefix-rft-no-clip is that in this varaint all N=8 rollouts are sampled from the same prefix (similar to original DR-PO paper). This results in the poor performance (33.8), likely due to the severe distribution shift between the guided training phase and the unguided inference phase.

Table 5: Comprehensive ablation results on Qwen2.5-Math-7B. Prefix-RFT outperforms all variants, justifying the synergy of prefix guidance, dynamic advantage weighting, and entropy-based clipping.

| Method Variant | MATH | Olympiad | Minerva | AIME 24 | AIME 25 | AMC | Avg. |
|---|---|---|---|---|---|---|---|
| *Prefix Gradient Strategies* | | | | | | | |
| Prefix-RFT (Freeze Prefix / All-Clip) | 84.8 | 48.3 | 37.9 | 24.1 | 16.4 | 61.2 | 45.4 |
| Prefix-RFT (Update All / No-Clip) | 83.2 | 49.8 | 41.5 | 22.5 | 18.5 | 58.5 | 45.7 |
| *Method Variants* | | | | | | | |
| Static Weight (0.001) | 83.2 | 47.6 | 39.0 | 26.0 | 18.9 | 58.4 | 43.8 |
| RFT + On-policy Clip | 85.0 | 45.0 | 39.3 | 20.2 | 12.7 | 60.6 | 43.8 |
| DR-PO Variant | 66.8 | 34.4 | 34.2 | 13.8 | 10.7 | 42.8 | 33.8 |
| **Prefix-RFT (Ours)** | **88.4** | **55.7** | **40.3** | **31.8** | **26.4** | **68.2** | **51.8** |

**Gradient Magnitude Analysis** To justify the motivation of our entropy-based clipping strategy, we analyze the gradient magnitudes derived from off-policy demonstration tokens compared to on-policy RFT tokens. As discussed in Sec. 3, because the offline expert policy $\pi_{\text{off}}$ (e.g., DeepSeek-R1) may be distributionally distant from the current policy $\pi_\theta$, the model often assigns low probabilities to demonstration tokens, resulting in large gradients for the log-likelihood objective. We compare the average gradient norms of four training methods on Qwen2.5-Math-7B: (1) **SFT-R1-trace** (pure SFT on the demonstrations), (2) **Prefix-RFT (no-clip)** which updates all prefix tokens without clipping, (3) **Prefix-RFT (all-clip)** which freezes the prefix (gradient = 0), and (4) **Standard RFT**. As shown in Table 6, the gradients from the full SFT trace are orders of magnitude larger than RFT gradients. More importantly, *Prefix-RFT (no-clip)* exhibits gradient norms nearly double those of *Prefix-RFT (all-clip)* and *RFT*, despite prefix tokens constituting only a small fraction ($5\% - 10\%$) of the total tokens in the batch. This confirms that a small number of off-policy tokens can disproportionately dominate the optimization landscape if left unconstrained, leading to instability (e.g., response length explosion) as observed in our ablation studies.

Table 6: Comparison of gradient norms across different training stages. Results show that unclipped off-policy prefixes generate significantly larger gradients than on-policy rollouts.

| Method | 0-25 steps | 25-50 steps | 50-100 steps | 100-200 steps |
|---|---|---|---|---|
| SFT (R1 Traces) | 5.31 | 2.01 | 1.61 | 1.78 |
| Prefix-RFT (zero-clip) | 0.44 | 0.27 | 0.27 | 0.27 |
| Prefix-RFT (no-clip) | 0.23 | 0.13 | 0.13 | 0.13 |
| RFT | 0.18 | 0.12 | 0.12 | 0.12 |

**Statistical Significance Analysis** To assess the robustness of our reported improvements and ensure they are not artifacts of random seed selection, we conducted 5 independent inference runs for both Prefix-RFT and the strongest baseline, LUFFY (checkpoint released from their paper). We report the mean and standard deviation across these runs in Table 7. The results demonstrate that Prefix-RFT consistently outperforms LUFFY across all benchmarks with tight confidence intervals. For instance, on AIME 2025, we observe a substantial margin of +4.07% (25.13 vs 21.06), confirming the statistical significance of our contribution.

Table 7: Comparing Prefix-RFT against the strongest concurrent baseline, LUFFY with mutiple inference runs

| Method | MATH | Olympiad | Minerva | AIME 2024 | AIME 2025 | AMC | Avg. |
|---|---|---|---|---|---|---|---|
| LUFFY | $87.16 \pm 0.83$ | $54.28 \pm 0.51$ | $38.09 \pm 1.74$ | $28.46 \pm 0.45$ | $21.06 \pm 0.78$ | $66.17 \pm 0.41$ | 49.20 |
| **Prefix-RFT** | **$87.60 \pm 0.58$** | **$56.33 \pm 0.52$** | **$39.97 \pm 0.45$** | **$31.88 \pm 0.41$** | **$25.13 \pm 0.96$** | **$68.18 \pm 0.11$** | **51.52** |

## A.4 PRINCIPLED DESIGN AND MANAGEMENT OF HYPERPARAMETERS

Here we want to once again highlight the principle for introducing the entropy clipping and decay machanism, and link them to the aforementioned conclusions of ablation studies. Overall, these components are grounded in two core principles: *constrained and targeted optimization on off-policy data* and *position bias & curriculum learning*. Here, we outline the rationale behind these designs and provide guidelines for practitioners.

**Entropy-based Clipping for Targeted Learning:** The primary role of clipping is to manage the huge distribution gap between the offline expert ($\pi_{\text{off}}$) and the current policy ($\pi_\theta$). As shown in Table 6, gradients from offline tokens can be orders of magnitude larger than on-policy gradients. Without constraint, the model could quickly fit to superficial features of the demonstration (e.g., response length) rather than learning from both signals. We specifically target *high-entropy* tokens in this case because high entropy indicates model uncertainty, where the policy distribution is flat and the model is most likely to deviate from the expert path. Updating these tokens provides the richest learning signal. In contrast, low-entropy tokens usually represent confident predictions; updating them yields diminishing returns, as confirmed by our bottom 20% and random 20% ablation in Sec. 6.

**Cosine Decay for Position Bias and Curriculum Learning:** The decay scheduler serves two purposes: mitigating position bias and creating a natural training curriculum. Due to the autoregressive nature of LLMs, tokens at the end of a sequence are sampled less frequently. By starting with long prefixes and decaying to short ones, we ensure the model to see expert's actions in different positions. This also effectively creates a smooth transition from SFT (more guidance) to RFT (autonomous exploration), mirroring the standard two-stage post-training pipeline but within a unified loop.

These hyperparameters offer interpretable control knobs for practitioners. They can be managed by monitoring simple training metrics. For example, in our case, we monitored the rollout length: If the model begins generating excessively long or non-terminating sequences it is probably that the learning singal from R1 long cot demonstration is too strong. In this case, we just *increase* the clipping strength (reduce the ratio, e.g., 50% to 20%) or *decrease* the prefix length to reweight the training signal. We found our default settings (20% clip, cosine decay from 0.95 length ratio) to be robust across models we experiment with.

## A.5 LLM USAGE STATEMENT

This study leveraged a Large Language Model (LLM) for assistance in refining writing, including language polishing and grammatical correction, as well as for editing the code used to visualize experimental results. The ideation and all other aspects of the research were carried out by the author(s).

