# OpenReview forum: "Blending Supervised and Reinforcement Fine-Tuning with Prefix Sampling"
_ICLR.cc/2026/Conference — Submitted to ICLR 2026_

### Official Review · Reviewer_k1KL · 2025-10-20

**Soundness:** 2
**Presentation:** 2
**Contribution:** 2
**Rating:** 4
**Confidence:** 3

**Summary:**

This paper introduces Prefix-RFT, a hybrid approach that blends SFT and RL for LLM post-training. In particular, the method randomly samples a prefix from each expert demonstration and lets the actor model generate the continuation. This hybrid sequence, including off-policy prefix and on-policy continuation, is then used as a rollout sample in the RFT update. To ensure stability, the method applies entropy-based clipping on off-policy prefix tokens and uses a cosine decay scheduler to shorten prefix length over time, creating a curriculum from SFT to RFT. The experiments exhibit superior performance compared to SFT, RFT, and SFT-then-RFT baselines on math and reasoning benchmarks.

**Strengths:**

1. The topic of bridging SFT and RL for better generalized LLM reasoning is significant.

2. Experiments show promising results compared to baselines.

3. Extensive analyses are provided to support the validity of the method.

**Weaknesses:**

1. The presentation can be improved, especially the notations for introducing the method. For example, in Eq. (1), I don't understand what $\sum_{y_t^i\sim\pi_{\theta_{old}}}$ means, but to guess it might be the sum over $t$. In Eq. (2), again, the subscript is confusing, and it is hard to tell whether the prefix tokens from demonstrations contribute to the gradient from the formula.

2. The paper combines tricks like entropy-based clipping, which can bring notable improvement to RFT on its own [1]. It is questionable whether the promising results come from the tricks or the proposed hybrid approach of Prefix-RFT.

3. There are too many tunable configurations, including entropy clipping ratio, prefix sampling decay scheduler, etc, and they do not seem to be robust. For example, Figure 4 shows that simply changing the clipping ratio to 50% hugely degrades the performance from 50% to <45%. Figure 5 shows that the sampling schedule can also significantly affect the results. These non-robust configurations can bring a huge burden to practitioners.

[1] Wang, Shenzhi, et al. "Beyond the 80/20 rule: High-entropy minority tokens drive effective reinforcement learning for llm reasoning." arXiv preprint arXiv:2506.01939 (2025).

**Questions:**

1. What is the principle behind the cosine decay scheduler? Can it be any decaying scheduler, e.g., linear?

2. Is it possible to conduct additional experiments on RFT with entropy-clipping for ablation?

---

> ### Author Response · Authors · 2025-11-20
> **Response to reviewer k1KL [1/n]**
>
> > **The presentation can be improved, especially the notations for introducing the method. For example, in Eq. (1), I don't understand what it means, but to guess it might be the sum over. In Eq. (2), again, the subscript is confusing, and it is hard to tell whether the prefix tokens from demonstrations contribute to the gradient from the formula.**
>
> We thank the reviewer for pointing this out. We have improved the presentation and the explanation of Eqs. 1 and 2 in the revised manuscript to ensure clarity. Additionally, please see our direct clarification of these two equations below.
>
> - **Eq. 1 (general objective)**: After showing in Section 2 that SFT and RFT gradients are both weighted log-probability updates, Equation (1) is introduced to present a **general, abstract learning objective** for a hybrid post-training paradigm. The first term (weighted by $\alpha$) represents "learning from exploration" (tokens from on-policy rollouts); the second term (weighted by $\beta$) represents "learning from imitation" (tokens from offline demonstrations). The main idea is that for any training batch, its gradient can be induced from tokens sampled from different distributions: (1) the model itself ($\pi_{\theta_{old}}$); and (2) an off-policy expert distribution (could be human experts or a superior teacher model, denoted as $\pi_{off}$).
> - **Eq. 2 (Specific instantiation)**: Our method (Prefix-RFT) is a specific **instantiation of Eq. 1** where "imitation" tokens are from off-policy prefixes, and "exploration" tokens are from on-policy continuations and standard RFT rollouts. To answer the reviewer's direct question: Yes, the prefix tokens absolutely contribute to the gradient (represented by the second term in our formulation). We hope this explanation resolves the confusion.
>
> > **The paper combines tricks like entropy-based clipping, which can bring notable improvement to RFT on its own [1]. It is questionable whether the promising results come from the tricks or the proposed hybrid approach of Prefix-RFT.**
>
> We thank the reviewer for referring us to this work; we were already aware of it and have cited it in our submission. We also followed the reviewer's suggestion and ran the "RFT + Entropy Clipping" baseline (i.e., standard RFT with our "top 20%" clipping applied to its *on-policy* gradients). The results are as follows:
>
> | method         | Math500 | Olympiad | minerva | aime24 | aime25 | amc   | Average |
> | -------------- | ------- | -------- | ------- | ------ | ------ | ----- | ------- |
> | On policy—clip | 85.0    | 45.0     | 39.3    | 20.23  | 12.71  | 60.59 | 43.8    |
> | prefix-rft     | 88.4    | 55.7     | 40.3    | 31.8   | 26.4   | 68.2  | 51.8    |
>
> According to the additional result, the method performs similarly to the naive RFT (45.5, see Table 1), demonstrating that the large performance gap is closed by our core contribution: explicitly reinforcing high-quality expert prefixes as part of the RFT update. Furthermore, we highlight that the clipping in our work is applied differently from that of [1]. They apply a mask to the entire batch to focus RFT on critical *on-policy* tokens. Our method applies a mask only to the off-policy prefix tokens. All other on-policy tokens are updated as usual.
>
> In short, clipping is for stabilization, while reinforcing the prefix is the core driver of performance. We will add this new ablation and clarification to the revised paper (Appendix, Table 3).

---

> ### Author Response · Authors · 2025-11-20
> **Response to reviewer k1KL [2/n]**
>
> > **There are too many tunable configurations, including entropy clipping ratio, prefix sampling decay scheduler, etc, and they do not seem to be robust. For example, Figure 4 shows that simply changing the clipping ratio to 50% hugely degrades the performance from 50% to <45%. Figure 5 shows that the sampling schedule can also significantly affect the results. These non-robust configurations can bring a huge burden to practitioners.**
>
> To highlight the robustness of our method,  we first pinpoint the consistent gains across multiple models and datasets and present additional results on Qwen3-1.7B-base. Then we compare our method with other concurrent works to show that our configuration is more intuitive and manageable.
>
> **Consistency across multiple models and datasets:** We have validated these exact configurations across three distinct models—**Qwen2.5-Math-7B, Qwen2.5-Math-1.5B, and Llama-3.1-8B**—and observed consistent improvements (Tab. 1&2). We also demponstrate that the proposed recipe transfers across different data sizes and demonstration qualities (Tab. 3). For further validation, we also experiment with the **Qwen3-1.7B model base**. The results indicate that our method also outperforms the SFT and RFT baselines with the same configuration:
>
> | Method     | AIME24 | AIME25 | AMC  | MATH-500 | Minerva | Olympiad | Avg. |
> | ---------- | ------ | ------ | ---- | -------- | ------- | -------- | ---- |
> | SFT        | 10.0   | 10.4   | 34.5 | 66.6     | 24.6    | 27.2     | 28.9 |
> | RFT        | 11.6   | 5.4    | 35.4 | 68.4     | 30.1    | 31.9     | 30.3 |
> | Prefix-RFT | 10.0   | 8.1    | 36.3 | 74.2     | 32.7    | 35.3     | 32.8 |
>
> To highlight the robustness of the introduced method, we append the results above at the end of Sec. 6 (lines 470-526).
>
> **Comparison to Concurrent Work (LUFFY and UFT) regarding tunable hyperparameters:** We also want to highlight how our concurrent works, LUFFY and UFT, control their off-policy updates:
>
> - LUFFY introduces a policy reshape function $f(x) = \frac{x}{x+\gamma}$. This reweights the gradient, pushing the model to focus on tokens with a probability close to $\gamma$. They set $\gamma$ as 0.1 in experiments.
> - UFT puts a very small, fixed weight (0.001) on the prefix.
>
> According to LUFFY's paper (Figure 9, App. E.4), its performance drops significantly when $\gamma$ varies from 0.1 to 0.05. A similar fluctuation can also be observed for UFT (Tab. 3 and 4, App. B.3) when tuning the weight across [0.0005, 0.001, 0.002].
>
> **Interpretability** Compared to these weight-based methods, our configuration is more intuitive and manageable. Here is the principle of introducing these two configurations:
>
> 1. **A constrained update from off-policy data.** Sometimes the demonstration prefix can be highly off-policy. As Fig. 4 shows, using the random 20% variant provides a "too sharp" update to the training (e.g., the model starts to mimic the demonstration's length). Thus, the update from off-policy data should be carefully constrained. We prioritize high-entropy tokens because the model’s own prediction on these tokens is uncertain (the distribution is relatively flat), and that’s precisely where the model may deviate from expected behavior and where reinforcement is needed. As Fig. 4 also shows, the bottom 20% (low-entropy) variant learns very little, further consolidating this point.
> 2. **Addressing positional bias and creating an SFT-to-RFT curriculum.** Owing to the autoregressive nature, tokens in the suffix are naturally sampled less often. We thus introduce a decay mechanism to address this. Meanwhile, the longer the prefix, the more information the model learns from it. So this decay also creates a curriculum from SFT to RFT, which corresponds to the practical two-stage training pipeline. Note that while the Uniform scheduler underperforms the Cosine Decay scheduler, it is still **significantly better** than pure RFT (+3.2) and SFT (+4.6), showing the core benefit of the prefix.
>
> Given these two principles, the clip ratio and decay scheduler basically control **how much information is taken from the prefix and where.** We believe this is not a "non-robust" setting but a direct, interpretable control for practitioners based on their prior knowledge about the model and data.
>
> Meanwhile, we can also manage them by monitoring training dynamics. For example, in our case, if we observe the model generating overly long or non-stopping rollouts (owing to learning from R1 with long CoT), we can increase the clip_ratio to reweight the training signal. For another example, if we see that the model’s reward with the prefix quickly drops with the decay, we can tell that the model needs more information from the prefix and can thus slow the decay. We have included this discussion in our revised manuscript to highlight how to adjust the two mechanisms (Appendix A.4).

---

> ### Author Response · Authors · 2025-11-25
> **Gentle Reminder for feedback**
>
> Dear Reviewer k1KL,
>
> We sincerely appreciate your constructive review.
>
> As the discussion period is expected to conclude next week, could you please let us know whether our response addresses your concerns?  We are happy to answer any further questions or provide necessary clarifications or results to resolve your doubts.
>
> Thank you,
>
> The Authors

---

> ### Author Response · Authors · 2025-11-30
> **Response to reviewer k1KL [3/n]**
>
> > **What is the principle behind the cosine decay scheduler? Can it be any decaying scheduler, e.g., linear?**
>
> The principle behind the decay is explained in the previous question. We choose cosine decay because it is widely adopted. We have not tried linear decay, but we believe that it will achieve a similar performance.

---

### Official Review · Reviewer_Q8Mt · 2025-10-28

**Soundness:** 3
**Presentation:** 4
**Contribution:** 2
**Rating:** 4
**Confidence:** 3

**Summary:**

This paper proposes a finetuning method for language models that combines supervised finetuning (SFT) with reinforcement finetuning (RFT). The proposed method, called Prefix-RFT, initializes the on-policy rollouts of RFT from prefixes of existing demonstrations. The idea is to guide RFT toward exploring more favorable outputs, while maintaining the goal-oriented objective of RFT. Empirical evaluations over math reasoning datasets show that Prefix-RFT outperforms SFT, RFT, and several other finetuning approaches.

**Strengths:**

1. The presentation of Prefix-RFT is clear. More broadly, I find the paper to be well-written and relatively easy to follow.

2. How to best combine RFT with demonstrations is an active and important area of research, toward identifying robust language model finetuning guidelines.

3. The empirical analysis in Section 5 helps shed light on the mechanisms underlying Prefix-RFT and its relation to SFT and RFT.

**Weaknesses:**

1. Unfortunately, it seems that the main technical innovation of Prefix-RFT is not new. Aside from the arguably concurrent UFT paper [1], there is an even earlier work proposing to use demonstration prefixes to seed rollouts in policy gradient [2]. These methods are nearly identical, differing mostly in their technical details (e.g., amount of non-prefixed rollouts used in a batch and entropy-based clipping). While the paper does briefly mention [1], it is missing a reference to [2]. Given the vast similarity between the method proposed in [2] and Prefix-RFT, I believe it is necessary to clearly discuss the relation between these two works.

2. In light of [2], the main contribution of this paper is to show that the existing idea of using prefixes of demonstrations, along with a few other heuristics (e.g., entropy clipping) can work well for math reasoning datasets. This can be a worthy contribution if the method is shown to perform significantly better than alternatives. However, since the paper does not report standard deviation across random seeds (or any other measure of statistical significance), it is difficult to assess how robust are the performance gains reported in Table 1.



Review Summary and Recommendation
---

Overall, the idea behind Prefix-RFT is intuitive and clearly presented. However, since the core component of Prefix-RFT is not new, I find the contributions of this paper beyond existing works to be rather incremental. I therefore believe that it falls on the borderline. My current rating tends toward rejection, yet I am willing reconsider my assessment if the authors address the relation of Prefix-RFT to [2] and provide evidence that the gaps in performance in Table 1 are significant (e.g., what is the standard deviation of these results across random seeds for training?).


Additional (More Minor) Comments
---

- Typo in line 36: "it's"  should probably be "It is".

- Typo in line 56: Period after reference to Liu et al. 2025d should probably be a comma.



[1] Liu, M., Farina, G., & Ozdaglar, A. (2025). UFT: Unifying Supervised and Reinforcement Fine-Tuning. arXiv preprint arXiv:2505.16984.

[2] Chang, J. D., Zhan, W., Oertell, O., Brantley, K., Misra, D., Lee, J. D., & Sun, W. (2024). Dataset reset policy optimization for rlhf. arXiv preprint arXiv:2404.08495.

**Questions:**

--

---

> ### Author Response · Authors · 2025-11-20
> **Response to reviewer Q8Mt**
>
> We thank the reviewer for the positive comments on the clarity of our work. We address your concerns regarding the relation to prior works (specifically DR-PO) and the significance of the results as follows.
>
> > **The relation to the two mentioned related works.**
>
> First, we respectfully remind the reviewer that according to the ICLR 2026 reviewer guideline, “the authors are **not required to** compare to papers solely on arXiv.” The two papers mentioned, UFT and DR-PO, seem not to have been published before the submission. UFT, as noted in the **General Comment**, is a concurrent work. More importantly, we have provided a direct experimental comparison showing that UFT underperforms the naive RFT baseline in our setting, whereas Prefix-RFT achieves significant gains. The **DR-PO** work is indeed related. Here is the detailed comparison with DR-PO.
>
> **Differences in methodology, motivation, and experiment setting. **Here is the summary of the similarities and the differences:
>
> - **Similarity:** Both methods start from a partial offline datapoint and sample the continuation.
> - **Differences:** Aside from the difference in settings and studied domains, the most significant difference is that DR-PO does not count the loss on prefix tokens and thus does not blend SFT and RFT into a unified learning approach (this is our core motivation). As shown in the DR-PO paper, the RFT process is also started from an SFTed model. This may be why DR-PO does not need to learn from the prefix, because the model is already trained on it.
>
> We would also like to provide additional ablation results to **show why learning from prefixes is essential in our case**. To be specific, we supplement two variants:
> - **prefix-rft-allclip**: Based on the proposed Prefix-RFT method, we clip all gradients from prefix tokens (no loss from prefix tokens).
> - **DR-PO-variant**: We randomly sample a prefix from the demonstration data and perform RFT training with the continuation. Similar to the way proposed in DR-PO.
>
> The results are presented in the following table:
>
> | method              | Math | Olympiad | minerva | aime24 | aime25 | amc  | Avg  |
> | ------------------- | ---- | -------- | ------- | ------ | ------ | ---- | ---- |
> | prefix-rft-all-clip | 84.8 | 48.3     | 37.9    | 24.1   | 16.4   | 61.2 | 45.4 |
> | DR-PO variant       | 66.8 | 34.4     | 34.2    | 13.8   | 10.7   | 42.8 | 33.8 |
> | prefix-rft          | 88.4 | 55.7     | 40.3    | 31.8   | 26.4   | 68.2 | 51.8 |
>
> We observed that all_clip behaves similarly to pure RFT regarding both learning dynamics and performance. The DR-PO variant’s training reward is significantly higher, as its rollouts are from a high-quality prefix. However, there is a significant distribution gap between the training and test, leading to poor performance.
>
> To sum up, despite starting from a sampled prefix, our method also learns from this prefix to blend the SFT into the RFT training. And by playing with the lengths and numbers of the prefix, the proposed method could be regarded as an interpolation of SFT and RFT. Based on this framework, we introduce the entropy clip as a simple yet effective solution to address the off-policy nature of prefix tokens. We hope our discussions and additional experiments resolve your concern.
>
> The discussion is added to the revised manuscript. Related work is added to Appendix A.1, lines 678-680, and the additional results are in Table 3 (Appendix).
>
> > **Evidence of the performance gaps**
>
> Owing to time and computational constraints, we report the average result across independent inference runs. As we already achieved **a +6.3 / +7.7 / +7.6 / +4.0** performance gain against our baseline **RFT / SFT** and concurrent works **UFT / ReLIFT**, we compare with our strongest baseline, **LUFFY**, using five independent inferences and report the average and std.
>
> |                | MATH              | Olympiad          | Minerva           | AIME24            | AIME25            | AMC               | Avg        |
> | -------------- | ----------------- | ----------------- | ----------------- | ----------------- | ----------------- | ----------------- | ---------- |
> |LUFFY| 87.16±0.83      | 54.28±0.51      | 38.09±1.74      | 28.46±0.45      | 21.06±0.78      | 66.17±0.41      | 49.20      |
> |Prefix-RFT|**87.60±0.58** | **56.33±0.52** | **39.97±0.45** | **31.88±0.41 ** | **25.13±0.96 ** | **68.18±0.11 ** | **51.52 ** |
>
> According to the table, Prefix-RFT achieves a clear performance gain over LUFFY (e.g., +4.0% on AIME25) that is well outside the margin of error. Furthermore, as shown in Figure 6 (Appendix), we observe consistent gains across three different models, providing strong evidence of the method's robustness. For example, on Qwen2.5-Math-1.5B, we achieve a performance gain of about **+9.1 (+11.0)** compared with naive **SFT (RFT)**, as well as **+3.0** against the strong concurrent work **LUFFY**.
>
> We have revised the manuscript for **Minors corrections**

---

> > ### Comment · Reviewer_Q8Mt · 2025-11-23
> >
> > Thank you for the response and additional experiments. I went over them and the other reviews carefully.
> >
> > Regarding concurrent work. While [1] is very recent, [2] has been on arXiv since April 2024. The ICLR 2026 reviewer guidelines permit not comparing to unpublished arXiv papers, yet it is unclear whether it should be considered fair or best practice to ignore such papers completely. This is why I chose to mention it in the initial review. In line with the reviewer guidelines, I strongly encourage discussing the relation to [2], but will not view an exclusion of such a discussion as grounds for rejection.
> >
> > Regarding significance of empirical evaluation. I acknowledge that computational constraints may limit the ability to report standard deviations across training runs. I still believe that this limits the significance of the results and would recommend having multiple random seeds for training in at least in some settings (e.g., over a smaller subset of the data or using a smaller model compared to the setting of Table 1). Nonetheless, in light of the additional results with different random seeds for evaluation and the revisions to the manuscript, I am increasing my initial rating.
> >
> > [1] Liu, M., Farina, G., & Ozdaglar, A. (2025). UFT: Unifying Supervised and Reinforcement Fine-Tuning. arXiv preprint arXiv:2505.16984.
> >
> > [2] Chang, J. D., Zhan, W., Oertell, O., Brantley, K., Misra, D., Lee, J. D., & Sun, W. (2024). Dataset reset policy optimization for rlhf. arXiv preprint arXiv:2404.08495.

---

### Official Review · Reviewer_SL41 · 2025-10-30

**Soundness:** 2
**Presentation:** 4
**Contribution:** 2
**Rating:** 2
**Confidence:** 4

**Summary:**

This paper introduces Prefix Reinforcement Fine-Tuning (Pre-FT), a hybrid post-training method that leverages offline prefixes to guide exploration during reinforcement fine-tuning (RFT). By incorporating prefix guidance, Pre-FT expands an LLM’s knowledge boundary beyond that of RFT while mitigating the overfitting and memorization issues observed in supervised fine-tuning (SFT). Experimental results show that Pre-FT outperforms SFT, RFT, and other mixed-policy RFT approaches, and is robust to variations in both the quality and quantity of demonstration data.

**Strengths:**

- Empirical results show that Pre-FT consistently outperforms SFT, RFT, and other mixed-policy RFT methods.
- Pre-FT demonstrates strong robustness across varying quantities and qualities of demonstration data.
- The paper is well-written.

**Weaknesses:**

- **Lack of novelty and incremental contribution:** My main concern is that the use of off-policy data to enhance model capabilities has already been explored in several prior works [1, 2, 3]. Moreover, Pre-FT closely resembles UFT [3], which also applies the SFT loss to prefixes and uses RFT for partial continuations. The additional heuristics, such as entropy-based clipping and the cosine decay scheduler, appear too incremental to me.
- **Pass@1 does not reflect the reasoning capabilities of Pre-FT:** The paper reports only Pass@1 to evaluate the effectiveness of Pre-FT and other approaches. However, Pass@1 alone does not adequately capture whether Pre-FT expands reasoning capabilities beyond SFT and RFT methods [5].
## References
[1] Learning to Reason under Off-Policy Guidance. arxiv 2504.14945.

[2] Learning what reinforcement learning can’t: Interleaved online fine-tuning for hardest questions. arXiv:2506.07527v1.

[3] Uft: Unifying supervised and reinforcement fine-tuning. arXiv:2505.16984.

[4] Safety Alignment Should Be Made More Than Just a Few Tokens Deep. ICLR 2025 Oral.

[5] Does Reinforcement Learning Really Incentivize Reasoning Capacity in LLMs Beyond the Base Model?. AI4Math@ICML25 Oral.

**Questions:**

- Why does Pre-FT, by using prefix tokens as additional guidance, help mitigate the language mixing problem in RFT? Are there any experiments demonstrating that Pre-FT exhibits less language mixing compared to RFT?
- It would be interesting to investigate whether RFT can mitigate the shallow alignment problem in safety alignment [4]. Specifically, by using non-refusal prefixes during Pre-FT fine-tuning, can Pre-FT achieve deeper alignment on later tokens?
- How does Pre-FT perform compared to SFT and RFT models when evaluated using Pass@$k$ with a large $k$ sampling budget?
- What is the effect of freezing the prefix (i.e., not updating it) and training only on the partial continuations, or alternatively, updating all tokens in the prefix? How do these variants compare to Pre-FT?
- While the paper claims that the gradients from offline demonstrations can be significantly larger than those from RFT, empirical evidence supporting this phenomenon is necessary — especially given that the number of offline prefix tokens is much smaller than the number of continuation tokens used for training. The paper should further explain why significantly larger gradients from offline demonstrations could lead to an unstable training process..
- Why not assign an additional weight to the offline prefix tokens instead of relying on entropy-based clipping?

---

> ### Author Response · Authors · 2025-11-20
> **Response to reviewer SL41 [1/n]**
>
> We thank the reviewer for the thorough review and valuable feedback. We are glad the reviewer found our empirical results strong, our method robust, and the paper well-written. We first want to address the primary concerns regarding novelty and evaluation, then respond to other questions with clarifications or additional experimental results.
>
> > **Lack of novelty and incremental contribution**
>
> We would like to clarify the position of our work relative to the cited references.
>
> - **Concurrent Work:** As noted in the **General Comment**, the mentioned works, **LUFFY** (Yan et al., 2025), **ReLIFT** (Ma et al., 2025), and **UFT** (Liu et al., 2025b) are all contemporaneous works according to the ICLR 2026 reviewer guide [https://iclr.cc/Conferences/2026/ReviewerGuide]. As they have not been published in a peer-reviewed venue for over two months. As far as we know, LUFFY and UFT have been accepted to NeurIPS 2025. The ReLIFT is also a submission to the ICLR 2026. Therefore, Prefix-RFT should be evaluated as a parallel-developed solution to a shared significant problem rather than incremental work.
> - **Comparison to UFT:** We explicitly implemented UFT in our setting (Qwen2.5-Math-7B model + R1 demonstrations). As shown in the table in the **General Comment**, UFT (static weighting) slightly underperforms the naive RFT baseline (44.2 vs 45.5) in this complex reasoning setting, whereas Prefix-RFT achieves significant gains (**51.8**). We also provided a detailed comparison of the technique and experiment setting in the **General Comment.**
>
> Therefore, apart from the idea of using prefixes similar to UFT, we also propose and validate a practical framework that could actually work with larger models, more difficult tasks, and more complex offline demonstrations. And our proposed design choices (entropy clipping and the decay scheduler) are not incremental additions but **rather essential, well-justified (as shown in our ablation studies) solutions** to the more difficult setting we tackle.

---

> > ### Author Response · Authors · 2025-11-20
> > **Response to reviewer SL41 [2/n]**
> >
> > > **Pass@1 does not reflect the reasoning capabilities of Prefix-FT & How does Pre-FT perform compared to SFT and RFT models when evaluated using Pass@ with a large sampling budget?**
> >
> > We respectfully argue that pass@1 is one of the standard, informative, and widely used metrics in this field. Meanwhile, we agree that it does not reveal the model’s capability ceiling. We thus follow the suggestion to use a large sampling budget to compare the base model, SFT, RFT, and Prefix-RFT. We set n=2048 to calculate pass@1 to pass@2048. We use the same inference setting as in our main experiments for all methods.
> >
> > Here are our results:
> >
> > AIME 24
> > |  | pass@1     | pass@2     | pass@4     | pass@8     | pass@16    | pass@32    | pass@64    | pass@128   | pass@256   | pass@512   | pass@1024  | pass@2048  |
> > | -------- | ---------- | ---------- | ---------- | ---------- | ---------- | ---------- | ---------- | ---------- | ---------- | ---------- | ---------- | ---------- |
> > | Base     | 0.1522     | 0.2362     | 0.3249     | 0.4059     | 0.4800     | 0.5503     | 0.6179     | 0.6811     | 0.7388     | 0.7921     | 0.8359     | 0.8667     |
> > | **SFT**  | 0.2505     | 0.3378     | 0.4360     | 0.5318     | 0.6095     | 0.6645     | 0.6967     | 0.7143     | 0.7293     | 0.7467     | 0.7620     | 0.7667     |
> > | **RFT**  | 0.2729     | 0.3442     | 0.4183     | 0.4891     | 0.5511     | 0.6023     | 0.6459     | 0.6898     | 0.7390     | 0.7948     | 0.8542     | **0.9000** |
> > | **PRFT** | **0.3124** | **0.4008** | **0.4893** | **0.5690** | **0.6324** | **0.6794** | **0.7164** | **0.7547** | **0.7992** | **0.8405** | **0.8728** | **0.9000** |
> >
> > AIME25
> > | | pass@1     | pass@2     | pass@4     | pass@8     | pass@16    | pass@32    | pass@64    | pass@128   | pass@256   | pass@512   | pass@1024  | pass@2048  |
> > | ---------- | ---------- | ---------- | ---------- | ---------- | ---------- | ---------- | ---------- | ---------- | ---------- | ---------- | ---------- | ---------- |
> > | Base       | 0.0713     | 0.1142     | 0.1668     | 0.2256     | 0.2889     | 0.3531     | 0.4164     | 0.4832     | 0.5561     | 0.6236     | 0.6727     | 0.7000     |
> > | SFT        | 0.2266     | 0.2902     | 0.3460     | 0.3970     | 0.4474     | 0.4952     | 0.5371     | 0.5732     | 0.6047     | 0.6310     | 0.6489     | 0.6667     |
> > | RFT        | 0.1215     | 0.1497     | 0.1843     | 0.2331     | 0.3036     | 0.3929     | 0.4831     | 0.5601     | 0.6163     | 0.6494     | 0.6745     | 0.7000     |
> > | Prefix-RFT | **0.2598** | **0.2977** | **0.3468** | **0.4007** | **0.4537** | **0.5078** | **0.5675** | **0.6274** | **0.6780** | **0.7195** | **0.7539** | **0.7667** |
> >
> > These experiments (added to our manuscript, Appendix A.3, Table 2) confirm that Prefix-RFT maintains its significant lead over SFT and RFT, demonstrating superior reasoning capabilities at a larger sampling budget.
> >
> > Note that the evaluation in our submission is *not limited* to a single-sampling metric. As shown in Table 1, for the more challenging benchmarks (AIME 2024/25, AMC), our primary reported metric is avg@32, a larger-sampling-budget metric. Prefix-RFT outperforms most baselines here. Furthermore, our analysis in Section 5.1 and Figure 2b uses Best@16 to explicitly demonstrate that Prefix-RFT effectively elevates the upper bound of the RFT tuning on problems that pure RFT finds unsolvable.
> >
> > > **Why does Pre-FT, by using prefix tokens as additional guidance, help mitigate the language mixing problem in RFT? Are there any experiments demonstrating that Pre-FT exhibits less language mixing compared to RFT?**
> >
> > We appreciate the opportunity to clarify this. To be precise, we do NOT claim that Prefix-RFT explicitly solves the language mixing issue. We introduced this concept solely in the introduction to illustrate a known fact that pure RFT with sparse reward signals in complex, multi-step tasks can lead to unexpected behaviors. To further motivate our main proposal: SFT complements RFT by providing dense process supervision.
> >
> > > **It would be interesting to investigate whether RFT can mitigate the shallow alignment problem in safety alignment [4]. Specifically, by using non-refusal prefixes during Pre-FT fine-tuning, can Pre-FT achieve deeper alignment on later tokens?**
> >
> > Thanks for pointing out this exciting direction. However, the experiments the reviewer requests are out of scope for this paper, and given the limited time available during the rebuttal, we *cannot* provide additional results in the setting. Our intuition on how Prefix-RFT could help: The Shallow Alignment paper identifies that alignment primarily adapts only the first few tokens. Prefix-RFT could address this by sampling prefixes of varying lengths from harmful demonstrations and forcing the model to optimize the continuation. This effectively implements the "safety recovery" strategy suggested by the authors. However, in this case, it is still unclear whether we need to reinforce (train on) the prefix.

---

> > > ### Author Response · Authors · 2025-11-20
> > > **Response to reviewer SL41 [3/n]**
> > >
> > > > **What is the effect of freezing the prefix (i.e., not updating it) and training only on the partial continuations, or updating all tokens in the prefix? How do these variants compare to Pre-FT?**
> > >
> > > In short, both variants achieve similar performance with RFT and largely lag behind the proposed prefix-rft.
> > >
> > > Here are the ablation results with the proposed variants (both trained for 500 steps on Qwen2.5-Math-7B).
> > >
> > > | method                              | Math     | Olympiad | minerva  | aime24   | aime25   | amc      | Avg      |
> > > | ----------------------------------- | -------- | -------- | -------- | -------- | -------- | -------- | -------- |
> > > | prefix-rft-all-clip (freeze prefix) | 84.8     | 48.3     | 37.9     | 24.1     | 16.4     | 61.2     | 45.4     |
> > > | prefix-rft-no-clip (update all)     | 83.2     | 49.8     | 41.5     | 22.5     | 18.5     | 58.5     | 45.7     |
> > > | prefix-rft                          | **88.4** | **55.7** | **40.3** | **31.8** | **26.4** | **68.2** | **51.8** |
> > >
> > > - prefix-rft-all-clip: do not update the prefix tokens
> > > - prefix-rft-no-clip: we update all prefix tokens
> > >
> > > Though they achieve similar performance, their behavior during training is very different.
> > >
> > > - prefix-rft-allp-clip behaves very similarly to naive RFT
> > > - prefix-rft-zero-clip: receiving gradients from all prefix tokens from the R1 trace, its response length quickly explodes to 5k tokens. The behavior is very similar to the “prefix-rft-top80%” in our ablation study (Page 9, Figure 4, left).
> > >
> > > > **While the paper claims that the gradients from offline demonstrations can be significantly larger than those from RFT, empirical evidence supporting this phenomenon is necessary — especially given that the number of offline prefix tokens is much smaller than the number of continuation tokens used for training. The paper should further explain why significantly larger gradients from offline demonstrations could lead to an unstable training process.**
> > >
> > > The reviewer is correct to ask for evidence. In our rebuttal message, we explicitly compare gradients across different optimization strategies to support our claim.
> > >
> > > **Gradient magnitude from off-policy data is larger**: As we state in Section 3, expert demonstrations ($\pi_{off}$) could be far from the current policy ($\pi_{\theta}$). This means the probability $\pi_{\theta}$ assigns to these demonstration tokens is generally low. The gradient of the log-probability for these very low-probability tokens can be larger than the gradients from high-probability on-policy tokens. We also provide empirical evidence to support our statement.
> > >
> > > As shown in the following table (results are gathered based on Qwen2.5-Math-7B), the gradients from high-quality, complex, offline R1 demonstrations can be substantially larger than those of RFT. Meanwhile, the prefix-rft-zero-clip is significantly larger than prefix-rft-all-clip and pure RFT, showing that only a small portion of off-policy tokens could overwhelmingly dominate the RFT training. (Note that for prefix-rft-zero-clip, the rollout response length already explodes to 5k tokens after 25 steps; after that, the off-policy tokens only count for about 5~10% in each training batch, but almost double the gradient norm.)
> > >
> > > | method               | 0-25 steps | 25-50 steps | 50-100 steps | 100-200 steps |
> > > | -------------------- | ---------- | ----------- | ------------ | ------------- |
> > > | SFT-R1-trace         | 5.31       | 2.01        | 1.61         | 1.78          |
> > > | prefix-rft-zero-clip | 0.44       | 0.27        | 0.27         | 0.27          |
> > > | prefix-rft-all-clip  | 0.23       | 0.13        | 0.13         | 0.13          |
> > > | RFT                  | 0.18       | 0.12        | 0.12         | 0.12          |
> > >
> > > These new analyses of the gradient norm are added to Appendix Table 4.
> > >
> > > **“Unstable training process”:** We apologize if this term causes any misunderstandings. We meant to say that, owing to the very imbalanced gradients, if we do not constrain the optimization from off-policy tokens, their gradients could dominate the optimization process. As in our setting, the model’s response length quickly explodes because our offline demonstrations are long COT traces. We have shown and explained this in our analysis section (Figure 4). When updating 80% of prefix tokens, the training response length quickly increases to 5k tokens, and the training reward and benchmark performance also do not steadily improve. A similar issue is observed with the prefix-rft-zero-clip. To improve clarity, we have removed the term “instability” from the method section (lines 204-205).

---

> > > > ### Author Response · Authors · 2025-11-20
> > > > **Response to reviewer SL41 [4/n]**
> > > >
> > > > > **Why not assign an additional weight to the offline prefix tokens instead of relying on entropy-based clipping?**
> > > >
> > > > We would like to address the reviewer’s concern from two perspectives: (1) the ablation result on the weight-based method; (2) the explanation of the advantage of using clipping compared with static weight.
> > > >
> > > > First, since the discussed weighted-based method is similar to the approach taken by the concurrent work UFT, which applies a small, static weight (0.001) on the prefix to reweight the training signal, similarly, we run the following ablation: instead of entropy-clipping, we apply a static weight (0.001, same as UFT) to the prefix tokens. The results are as follows:
> > > >
> > > > | method           | Math500 | Olympiad | minerva | aime24 | aime25 | amc  | Average |
> > > > | ---------------- | ------- | -------- | ------- | ------ | ------ | ---- | ------- |
> > > > | prft-const-0.001 | 83.2    | 47.6     | 39.0    | 26.0   | 18.9   | 58.4 | 43.8    |
> > > >
> > > > The performance (and the training dynamics) is very close to PURE RFT. This may be because the constant assigned (we take this hyperparameter from UFT) is relatively small, so that the model does not effectively learn from the prefix tokens. However, due to the time and computation constraints, we are unable to conduct a more extensive hyperparameter search for this weighted-base variant. These additional results are added to Table 3, Appendix A.3
> > > >
> > > > Despite better performance, our method has the following inherent advantage over the static weighted-based method:
> > > >
> > > > 1. **Adaptive weight**: note that our method *already* uses a dynamic, utility-based weight, the estimated advantage assigned to the prefix-augmented rollout, which can indicate the value of the prefix. We also empirically demonstrate that our method learns more from the prefix on more complex problems (Figure 3(b)).
> > > > 2. **Targeted learning**: We empirically demonstrate that high-entropy tokens provide richer learning signals and are more critical to update (top 20%, random 20%, and bottom 20% ablations, Figure 4), whereas the static, weight-based method assigns a uniform weight to all prefix tokens.
> > > > 3. **Interpretability and adjustability**: compared to an absolute value weight. The clipping ratio is more interpretable and easier to adjust. Fluctuation can be observed for UFT (Tables 3 and 4, Appendix B.3 in their paper) when tuning the weight across [0.0005, 0.001, 0.002]. In contrast, we find a clear trend when we change 20% to 50% and 80% (Figure 4, page 9 in our submission).
> > > >
> > > > We hope these clarifications and our new experimental results (which we will add to the final paper) will fully address the reviewer's concerns. We thank the reviewer again for their time and valuable feedback.

---

> ### Author Response · Authors · 2025-11-25
> **Gentle Reminder for feedback**
>
> Dear Reviewer SL41,
>
> We sincerely appreciate your constructive review.
>
> As the discussion period is expected to conclude next week, could you please let us know whether our response addresses your concerns?  We are happy to answer any further questions or provide necessary clarifications or results to resolve your doubts.
>
> Thank you,
>
> The Authors

---

### Official Review · Reviewer_XmVn · 2025-11-02

**Soundness:** 4
**Presentation:** 4
**Contribution:** 4
**Rating:** 8
**Confidence:** 4

**Summary:**

The paper introduces Prefix-RFT, a hybrid post-training method that integrates supervised fine-tuning (SFT) and reinforcement fine-tuning (RFT) by sampling a demonstration prefix and letting the current policy complete it; the stitched sequence is then optimized with a PPO-style objective so imitation (prefix) and exploration (continuation) are updated in one pass. It frames SFT and RFT under a unified “log-probability gradient” view and adds two practical stabilizers—entropy-based clipping that backpropagates only through high-uncertainty demo tokens and a cosine-decay schedule for prefix length—to prevent dominance of off-policy tokens and mitigate position bias. On math reasoning benchmarks (e.g., AIME, AMC, MATH-500, Minerva, Olympiad) and some general tasks, Prefix-RFT outperforms standalone SFT, RFT, the two-stage SFT→RFT recipe, and concurrent mixed-policy baselines across model sizes. Analyses show it particularly helps on problems where pure RFT stalls, nudging the model toward expert distributions without degenerating into behavior cloning and inducing an example-wise transition from imitation to exploration; ablations confirm the importance of entropy clipping and the scheduling strategy. Overall, the contribution is a simple, easily integrated recipe that operationalizes a principled blend of demonstration guidance and goal-oriented updates for LLM post-training.

**Strengths:**

The paper’s originality lies in both its conceptual unification of SFT and RFT under a common gradient view and its pragmatic “prefix sampling” mechanism that stitches an off-policy demonstration prefix to an on-policy continuation while retaining PPO-style stability, delivering a clean bridge between imitation and exploration.   Methodological quality is high: the experimental suite covers diverse math-reasoning benchmarks with clear protocols, compares against relevant baselines, and includes well-designed ablations showing that updating only high-entropy demonstration tokens avoids prefix-driven overfitting, while a cosine decay on prefix length improves training dynamics and final scores.     Clarity is excellent, with intuitive figures and a step-by-step objective that makes the approach easy to adopt.  Finally, the significance is strong: results suggest the method reliably improves reasoning over pure RFT or sequential SFT→RFT and encourages the community to treat post-training as a blended process rather than two disjoint stages.

**Weaknesses:**

The evidence centers on math with exact checker rewards, so robustness under noisier or heuristic feedback remains unclear. There are no non-verifiable math settings (e.g., LLM-graded rationale quality) to test behavior when correctness can’t be deterministically checked. The study also relies primarily on Qwen backbones; results on additional families (e.g., Llama-3) would better assess backbone generality.

**Questions:**

See details in the weakness section.

---

> ### Author Response · Authors · 2025-11-19
> **Response to reviewer XmVn**
>
> We gratefully thank the reviewer for recognizing the contribution of our work.
>
> Here is our response to the mentioned weakness.
>
> 1. We agree with the reviewer that the existing supporting evidence is in the RLVR setting, where a checker only verifies the final output, providing minimal supervision for the rationale. We are also very interested in understanding how the proposed method generalizes to other settings.
> 2. Regarding backbone models, we also include results for Llama-3-8B-base. It is mentioned at line 281 on page 6 and in Figure 6 of the Appendix. Experimental results on the Llama-3-8B-base also highlight the effectiveness of our method compared to pure SFT (+8.6), RFT (+4.0), and the concurrent work LUFFY (+1.3). The relevant part has been revised and highlighted in the manuscript (lines 280-283 and Table 5).

---

### Author Response · Authors · 2025-11-19
**General comment to reviewers [1/2]**

We thank all four reviewers for their time and their detailed, constructive feedback. We are very encouraged that the reviewers converged on several key strengths of our work.

- **Clear Presentation:** The paper was consistently praised for being well-written (**SL41**, **Q8Mt**), clear (**Q8Mt**), and easy to adopt (**XmVn**).

- **Important Topic:** Reviewers agreed that the problem of bridging SFT and RFT is significant (**k1KL,** **XmVn**) and an active and important area of research (**Q8Mt**).

- **Strong Empirical Results:** Our proposed method consistently outperforms baselines (**SL41**), shows promising results (**k1KL**), and reliably improves reasoning. (**XmVn**).

- **Insightful Analysis:** We also provided the analyses that shed light on the mechanisms of SFT and RFT (**Q8Mt**), support the validity of the method (**k1KL**), and are well-designed (**XmVn**).

We appreciate this positive feedback and will now address the common concern raised: **Novelty and Concurrent Work**

We thank reviewers SL41 and Q8Mt for raising the crucial question of novelty and positioning our work relative to other recent papers. This is the most important point to address, as it frames our entire contribution. In total, our submission should be assessed as parallel independent work according to the ICLR 2026 review guide. We then highlight the difference with our closest work, UFT (mentioned by both reviewers SL41 and Q8Mt), and present its performance in our setting.

**1. ICLR 2026 Policy on Concurrent Work**

First, we respectfully note that according to the official ICLR 2026 reviewer guide[https://iclr.cc/Conferences/2026/ReviewerGuide], papers are considered contemporaneous if published within the last two months in a peer-reviewed venue. The guide explicitly states that arXiv is not a peer-reviewed venue. Therefore, we believe Prefix-RFT should be evaluated as a parallel-developed solution to a shared problem, rather than incremental work based on the prior work. Reviewer SL41 mentioned LUFFY[1], ReLIFT[2], and UFT[3]. Reviewer Q8Mt mentioned UFT[3]. To the best of our knowledge,

- ReLIFT [2] is an ICLR 2026 submission (https://openreview.net/forum?id=LzCBLrNoyM). This work is currently under review at this same conference.

- LUFFY [1] and UFT [3] are Concurrent/Recent: These works were only recently accepted to NeurIPS 2025 (results released one week before the ICLR deadline). Meanwhile, we also released the preprint version in early July. As such, they fall within the standard window of concurrent work.

Meanwhile, we fully respect their contributions and, in the spirit of thoroughness, have already included them in our discussion and used them as baselines (see the last paragraph of Section 3, lines 243-250, and the additional discussion in Appendix A.1, lines 686-701). According to Table 1, our proposed Prefix-RFT **achieves better performance** than these concurrent works.

---

> ### Author Response · Authors · 2025-11-19
> **General comment to reviewers [2/2]**
>
> **2. Detailed Comparison with UFT (concurrent work)**
>
> While these works are contemporaneous, here we provide a detailed comparison with UFT [3], our closest concurrent work and mentioned by reviewers SL41 and Q8Mt, to highlight differences in methods, experimental settings, and performance.
>
> Technically:
>
> - As described, UFT applies RFT with a sampled prefix and puts a small, **static weight (0.001)** on all prefix tokens.
> - **Prefix-RFT (Ours)** is fundamentally different. The weight on prefix tokens is **dynamic**, determined by the **advantage** of the entire hybrid (prefix + continuation) trajectory. We empirically demonstrated that this dynamic weight enables the model to learn more from demonstration for more complex problems (shown in Figure 3.b). To perform constrained, targeted learning from off-policy demonstrations, we propose a simple yet effective entropy-based clipping strategy that limits updates to high-entropy tokens. Both mechanisms are verified with our ablation studies (Figure 4 and Figure 5).
>
> This technical difference is not an incremental "heuristic"; it is a **necessary solution** for the more challenging problem setting we tackle. Here is the comparison of the two papers’ experiment settings:
>
> - **Tasks and Demonstrations:** UFT experiments on tasks such as Countdown and logic puzzle games. They also experiment with the MATH dataset. Overall, the offline SFT examples they employed are **"primarily composed of a few sentences."** Whereas we test our method on competition-level math (AIME, Olympiad) using complex reasoning traces from DeepSeek-R1 that are **up to 8,192 tokens long**.
> - **Model scale:** UFT uses Qwen2.5-0.5B/1.5B/3B and Llama-3.2-1B/3B. In contrast, we study larger models (Qwen2.5-Math-7B and Llama3.1-8B-base).
>
> To further prove this, we have run the UFT method in our setting. Specifically, we use the same configuration as described in the UFT paper: 300 steps of hybrid training with decay in prefix length, followed by 200 steps of pure RFT training. A static weight of 0.001 is applied to the prefix.
>
> | method     | Math     | Olympiad | minerva  | aime24   | aime25   | amc      | Avg      |
> | ---------- | -------- | -------- | -------- | -------- | -------- | -------- | -------- |
> | RFT        | 84.4     | 46.8     | 39.3     | 25.1     | 15.3     | 62.0     | 45.5     |
> | UFT        | 83.8     | 51.6     | 33.8     | 20.8     | 16.5     | 58.8     | 44.2     |
> | Prefix-RFT | **88.4** | **55.7** | **40.3** | **31.8** | **26.4** | **68.2** | **51.8** |
>
> Results clearly show that the UFT underperforms the naive RFT baseline and lags behind the proposed RFT method. To highlight this, we have included a detailed comparison with concurrent work in Appendix A.1 and added UFT as a baseline (Table 1).
>
> Apart from this common concern, we also provide additional ablations and clarifications to address reviewers’ individual questions. All additional experimental results and explanations are incorporated into our revised manuscript and highlighted in navy blue.

---

### Author Response · Authors · 2025-11-30
**Author final comment (summary for rebuttal phase)**

Dear Area Chair,

We note the recent information leak incident on OpenReview and sincerely appreciate your time and effort in reviewing our paper further. In our final comment, we would like to provide a summary of the rebuttal phase.

**Reviewer SL41 (Score: 2)** (1) The reviewer's primary concern is that the use of off-policy data to enhance model capabilities has already been explored in several prior works [1, 2, 3]; (2) The reviewer argued that "Pass@1 alone does not adequately capture whether Pre-FT expands reasoning capabilities"; (3) The reviewer also asked for other ablation studies.

- **Our Response:**
  - We clarified that these are concurrent works under the ICLR policy: https://iclr.cc/Conferences/2026/ReviewerGuide. [1] and [3] were recently accepted to NeurIPS 2025, and [3] is still under review at ICLR 2026, so our work should be evaluated as a parallel work to these references according to the ICLR 2026 Reviewer Guide. **Furthermore, *we have discussed these concurrent works in our original submission and demonstrated our superiority over them***.
  - We provided the required gradient-norm analysis (Appendix Table 4), empirically demonstrating that gradients from off-policy data could be much larger than those from on-policy data and can overwhelm our method's substantial training.
  - We also provide extensive additional explanations, ablation studies, and results for the proposed variant to address reviewers' questions. Overall, these new results further strengthen the statement in our original submission. Please see our detailed response to reviewer SL41.
  -  **The reviewer did not provide further feedback.**


**Reviewer Q8Mt (Score: 4 $\rightarrow$ 6)** stated (1) it is "necessary to clearly discuss the relation" to prior work DR-PO [4] and (2) noted that "without standard deviation... it is difficult to assess how robust are the performance gains."

- **Our Response:**
  - We analytically and empirically differentiated our work from DR-PO (empirically showing that DR-PO fails in our setting).
  - We provided results with standard deviations across independent runs, demonstrating that our gains over the strongest baseline (LUFFY) are statistically significant.
  - **The reviewer acknowledged these additions and increased the rating.**


**Reviewer k1KL (Score: 4)** (1) suggested that the clarity of the method (Eq.1 and E1. 2) section could be improved; (2) found it"questionable whether the promising results come from the tricks entropy-based clipping or the proposed hybrid approach"; (3) and worried that the "non-robust configurations can bring a huge burden to practitioners."

- **Response:**
  - We revised Eqs. 1 and 2, along with their explanation, to improve clarity.
  - We directly ran the proposed ablation of RFT + Entropy Clipping, which performs similarly to the original RFT, confirming that it is the hybrid Prefix-RFT objective that drives the success.
  - We further highlighted/demonstrated robustness of the configuration by validating that it works effectively across four different base models (Tab.2): Qwen2.5-Math-1.5B/7B, Llama-3-8B, and Qwen3-1.7B (Qwen3 is newly added durring discussion phase) and various data scenarios (Tab.3). We also provide a discussion on why our configuration is more intuitive and manageable compared to other concurrent works [1,3].
  - **The reviewer did not provide further feedback**


**Reviewer XmVn (Score: 8)** suggested that "results on additional families (e.g., Llama-3) would better assess backbone generality."

- **Response**: We clarified that the results of experiments using Llama-3-8B-base are included in the appendix in our original submission (now moved to the end of the paper).

All new results, discussions, and explanations have been added to our submission and are marked in navy blue. In closing, we sincerely thank you once again for your time and effort, and we hope our summary provides you with a more complete understanding of our work and rebuttal response.

1] Learning to Reason under Off-Policy Guidance. arxiv 2504.14945.

[2] Learning what reinforcement learning can’t: Interleaved online fine-tuning for hardest questions. arXiv:2506.07527v1.

[3] Uft: Unifying supervised and reinforcement fine-tuning. arXiv:2505.16984.

[4] Dataset reset policy optimization for rlhf. arXiv preprint arXiv:2404.08495.

---

### Meta-Review · Area_Chair_WR1B · 2025-12-31

**Summary:**

This work proposes a post-training method that blends the SFT (expert-distilled) and RL objectives, with the aim of improving the stability and performance of RL training alone. The core idea is to train on trajectories where the prefix is sampled from a fixed expert-model, and the suffix is sampled from the policy model. This reduces to RL for the empty prefix, and SFT for the full prefix.
The authors find this method, when combined with several heuristics (clipping, schedulers), can improve performance on math benchmarks.

The reviewers appreciated the motivation of this work, to combine the benefits of SFT and RL.
Reviewers also noted that the paper was written clearly, the idea was intuitive, and the experiments seemed correct.
The main reviewer concern was re the novelty of the work. No reviewer championed the novelty of the idea, and some reviewers noted that nearly identical methods had been proposed in prior work. Moreover, the experimental results were not strong enough to overcome the novelty concerns. In particular, several reviewers were concerned that the experiments made heavy use of heuristics, which were not fully explored or justified. After reading the paper myself, I agree with these concerns.

Overall, this is a well-written paper with solid experiments. The authors have done a commendable job of doing additional experiments in the rebuttal period, to address reviewer concerns. Ultimately however, the paper does not meet the novelty bar for acceptance to ICLR.

**Reviewer Concerns:**

See above. Reviewer concerns about statistical significance of experiments were largely addressed. Concerns about sensitivity to hyperparameters were only partially addressed. Concerns about novelty were not adequately addressed.

Regarding comparisons to prior work: While comparisons to unpublished arxiv work is not technically *required* by ICLR, we agree with Reviewer Q8Mt that in the ML community, it is best-practice to compare to relevant papers which are already > 1 year old. Otherwise, claims of novelty are not useful to the community.

**Reviewer Scores:**

I expect Reviewer Q8Mt to increase score to 6.
All other reviewers I do not expect to change scores.

---

### Decision · Program_Chairs · 2026-01-26

Reject